# On-Device Collaborative Language Modeling
# via a Mixture of Generalists and Specialists

**Dongyang Fan** [* 1]  **Bettina Messmer** [* 1]  **Nikita Doikov** [1]  **Martin Jaggi** [1]

## Abstract

On-device LLMs have gained increasing attention for their ability to enhance privacy and provide a personalized user experience. To facilitate private learning with scarce data, Federated Learning has become a standard approach. However, it faces challenges such as computational resource heterogeneity and data heterogeneity among end users. We propose CoMiGS (**Co**llaborative learning with a **Mi**xture of **G**eneralists and **S**pecialists), the first approach to address both challenges. A key innovation of our method is the bi-level optimization formulation of the Mixture-of-Experts learning objective, where the router is optimized using a separate validation set to ensure alignment with the target distribution. We solve our objective with alternating minimization, for which we provide a theoretical analysis. Our method shares generalist experts across users while localizing a varying number of specialist experts, thereby adapting to users' computational resources and preserving privacy. Through extensive experiments, we show CoMiGS effectively balances general and personalized knowledge for each token generation. We demonstrate that CoMiGS remains robust against overfitting—due to the generalists' regularizing effect—while adapting to local data through specialist expertise. We open source our codebase for collaborative LLMs.

## 1. Introduction

Large Language Models (LLMs) have been showing great success serving as foundation models, evidenced by their capability to understand a wide range of tasks, such as ChatGPT (OpenAI, 2023), Claude (Anthropic, 2023), Gemini (DeepMind, 2023) and etc. However, cloud-based in-

[*]Equal contribution [1]EPFL, Switzerland. Correspondence to: Dongyang Fan <dongyang.fan@epfl.ch>.

*Proceedings of the 42$^{nd}$ International Conference on Machine Learning*, Vancouver, Canada. PMLR 267, 2025. Copyright 2025 by the author(s).

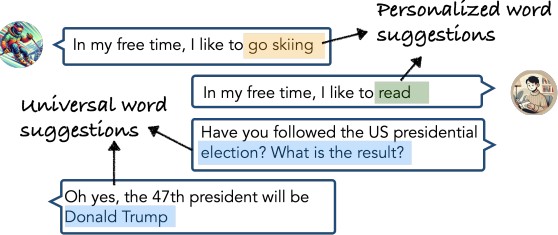

Figure 1: Chat box between two users with different characteristics. Next word prediction for smart keyboards should be tailored to users' topic preferences for personalization. However, to ensure factual accuracy and linguistic consistency, the results of next word prediction should maintain universality.

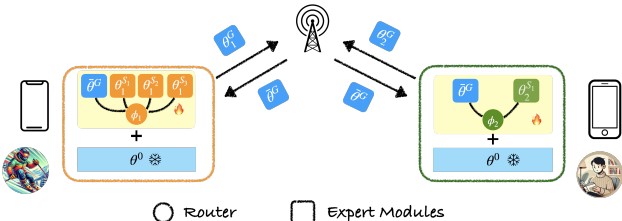

Figure 2: Diagram of our proposed method CoMiGS illustrated with a simplified 2-heterogenous-models setup (corresponding to the two users in Figure 1). Generalist experts ($\theta_1^G$, $\theta_2^G$) are aggregated across users, and specialist experts ($\{\theta_1^{S_i}\}_{i=1}^3$, $\{\theta_2^{S_1}\}$) and Routers ($\phi_1$, $\phi_2$) are kept local.

ference introduces significant delays for end users, and it often fails to meet their personalized needs (Ding et al., 2024; Iyengar & Adusumilli, 2024). Recently, there has been growing interest in deploying LLMs on edge devices, which offer benefits like lower latency, data localization, and more personalized user experiences (Xu et al., 2024). For instance, Apple (2024) recently launched on-device foundation models as part of its personal intelligence system. Meta (2024), Qwen (2024) newly released lightweight models with less than 3B parameters targeting edge AI.

On-device LLMs present challenges such as limited and variable computational resources, scarce and heterogeneous local data, and privacy concerns related to data sharing (Peng

et al., 2024; Wagner et al., 2024). Fine-tuning is typically performed on-device to quickly adapt to users' individual needs. While data sharing is a common solution to address local data scarcity, on-device data is often privacy-sensitive and must remain on the device. To overcome this, Federated Learning has been proposed as a method for enabling collaborative learning while preserving user privacy, allowing end users to collaborate by sharing model parameters (Chen et al., 2023; Zhang et al., 2023).

Federated fine-tuning of LLMs is predominately done through Low-Rank Adaptation (LoRA, Hu et al. (2021)) due to its lightweight nature so that the communication costs can be largely mitigated. Yet end devices may have different capacities, resulting in different LoRA ranks or different numbers of LoRA modules allowed on devices. Previous works have proposed various techniques for aggregating LoRA modules of different ranks (Cho et al., 2023; Bai et al., 2024). However, in both works, the devices are only equipped with shared knowledge, which makes the methods unsuitable when there is data heterogeneity across users. In such cases, a more personalized solution is needed.

End users' local data distributions can exhibit significant statistical heterogeneity. For instance, mobile device users may have distinct linguistic habits, topic preferences, or language usage patterns, leading to widely varying word distributions, as illustrated in different next word predictions to the same prompt "In my free time, I like to" in Fig. 1. As a result, personalized solutions are necessary. Wagner et al. (2024) explored three personalized collaborator selection protocols, allowing each end user to choose their collaborators. Although these protocols effectively address data heterogeneity, they depend on model aggregation, which can only occur when users share the same model architecture.

There has not yet been a solution to deal with both model heterogeneity and data heterogeneity. Towards this goal, we propose a novel **Co**llaborative learning approach via a **Mi**xture of **G**eneralists and **S**pecialists (CoMiGS). Our approach allows users to share part of the knowledge while keeping some knowledge user-specific, thus providing personalized solutions. We name the shared part *generalists* and the user-specific part *specialists*. Like all previous works, the generalists and specialists are simply LoRA modules. At the same time, as long as the shared part can be aggregated, the user-specific part can be of different sizes, which can be adapted to various device capacities, as illustrated by different numbers of specialists across users in Figure 2.

We integrate the expertise of generalists and specialists using a learned router that determines aggregation weights, following the Mixture-of-Experts (MoE) architecture (Fedus et al., 2022b). As in typical MoE designs for language modeling (Jiang et al., 2024; Fan et al., 2024), we also use tokens as the routing unit. Although users may have differ-

ent topic preferences or linguistic styles, they should still share common phrases, for example, when talking about factual knowledge (the result of the US presidential election should be a universal fact in Figure 1). Our goal is to route these shared tokens to the generalists so they can be jointly learned across users.

We further notice a hierarchical structure between the router and the experts: the router dynamically assigns tokens based on emerging expert specializations, while the experts refine their roles to optimize token processing under the router's guidance. Towards addressing this, we formulate our learning objective as a bi-level optimization problem and propose a new first-order algorithm based on alternating minimization as a solution. Our method enjoys convergence guarantees and is resource-efficient for deployment.

In summary, our contributions are as follows:

- We propose a novel approach (CoMiGS) for on-device personalized collaborative fine-tuning of LLMs. Key parts of our approach are: 1) an innovative bi-level formulation of the MoE learning objective (Section 3.2); 2) a new algorithm based on alternating minimization (Alg.1); 3) a theoretical analysis with a proof showing linear convergence rate under suitable assumptions (Section 3.4).

- Our collaborative framework effectively addresses both *data heterogeneity* (Section 4.2), concerning diverse local data distributions across users, and *computational resource heterogeneity* (Section 4.3), with respect to varying local model architectures, making it the first model to accomplish both.

- Our framework separates model heterogeneity from data quantity (Section 4.4). Users with larger local datasets benefit from a bigger model, while users with more powerful models but smaller datasets are less prone to overfitting.

- CoMiGS is resource-efficient: it adds marginal ($+1.25\%$) computational overhead and memory requirement compared to FedAvg, while reducing communication costs by 50% (Section 4.5).

- We release a codebase[1] for collaborative LLMs that allows users to easily implement various collaboration strategies, facilitating and advancing future research efforts in this field.

## 2. Related Work

**Collaborative Fine-Tuning for LLMs.** Recently, researchers have been investigating the application of Federated Learning in language tasks. Due to the substantial number of model parameters in LLMs, the research has largely targeted the stages following pre-training, often uti-

---

[1]Our code base is available at `https://github.com/epfml/CoMiGS`

lizing parameter-efficient techniques such as adapters. Mohtashami et al. (2023) explored a teacher-student social learning framework to aggregate private-sensitive instructions. Zhang et al. (2023) directly applied FedAvg (McMahan et al., 2017) to aggregate LoRA parameters during instruction tuning, and reported increased performance in downstream tasks. Following that, there are various works focusing on addressing resource heterogeneity where users are equipped with different LoRA ranks. HetLoRA (Cho et al., 2023) and FlexLoRA (Bai et al., 2024) provide different ways of aggregating and distributing LoRA modules of heterogenous ranks. However, these approaches are not designed to cope with heterogeneous data on device. In contrast, Sun et al. (2024) demonstrated that freezing LoRA A matrices at initialization leads to improved performance on heterogeneous data. Building on this, Guo et al. (2024) showed that consistently aggregating LoRA A matrices can yield even greater performance gains. Meanwhile, Wagner et al. (2024) introduced personalized approaches to effectively address data heterogeneity by employing three distinct collaborator selection mechanisms. However, these approaches require all users to utilize the same model architecture. Unlike previous works, our framework deals with both model heterogeneity and data heterogeneity. Moreover, our method offers personalized solutions at a token level, as opposed to the client-level approach in Wagner et al. (2024).

**Mixture of Global and Local Experts.** Gaspar & Seddon (2022) introduced a fusion of global and local experts for activity prediction based on molecular structures. Each local expert is tailored to a specific chemical series of interest using loss masking, while a global expert is trained across all series. Simultaneously, a routing network learns to assign soft merging scores. This approach yielded superior empirical results compared to single experts. Dai et al. (2024) developed DeepSeekMoE by deterministically assigning every token to "shared" experts, whereas "routed" experts are assigned tokens based on a learnable router. DeepSeekMoE is able to approach the upper bound performance for MoE models. For both works, the notion of shared/global is with respect to input samples, i.e. a shared/global expert should see all input samples. In a collaborative setup, FDLoRA (Qi et al., 2024) learns dual LoRA modules on each client to capture personalized and global knowledge respectively. Client-wise fusion weights are learned towards the end of training to combine the two sets of knowledge. In comparison, pFedMoE (Yi et al., 2024) jointly learns a shared homogeneous small feature extractor, a localized heterogeneous feature extractor, and a localized routing network in an end-to-end fashion, demonstrating strong performance in the vision domain. Our work is closely related to these two works, while introducing key innovations that adapt to model heterogeneity and allow for fine-grained adaptation to target distributions.

## 3. Method

We aim to improve personalized performance for each end user, through a fine-grained mixture of general and personalized knowledge. Building on the hierarchical insights of MoE learning, we formulate our learning objective into a bi-level optimization problem, where expert parameters are learned using the relatively large-sized training sets, while routing parameters are updated using the small-sized validation sets. We further let experts diversify into generalists and specialists via parameter aggregation or localization. As the problem solver, we provide a multi-round gradient-based algorithm, of which the pseudo codes are presented in Appendix A.

### 3.1. Notions and Problem Setup

Each user has a training set $X_i^{\text{train}}$, a small validation set $X_i^{\text{valid}}$ and a test set $X_i^{\text{test}}$, and the task is next token prediction. The validation set $X_i^{\text{valid}}$ and the test set $X_i^{\text{test}}$ are sampled from the same distribution $\mathcal{P}_i^{\text{target}}$ (note this is a fuzzy concept in the language domain, by the same distribution we mean from the same topic/category). The training set, $X_i^{\text{train}}$, can be sampled from a different distribution than $\mathcal{P}_i^{\text{target}}$. This is to address scenarios where distribution shifts may occur over time, such as changes in topics reflected in the typing data of mobile phone users. As illustrated in Figure 2, there are two sets of model parameters within each user: expert parameters, denoted as $\boldsymbol{\Theta} = \boldsymbol{\theta}^G \cup \{\boldsymbol{\theta}_i^S\}$, where $\boldsymbol{\theta}^G$ is shared across the users and $\{\boldsymbol{\theta}_i^S\}$ are user-specific specialist parameters; and routing parameters, denoted as $\boldsymbol{\Phi} = \{\boldsymbol{\phi}_i\}$. $i \in \{1, 2, .., N\}$ is the user index. We use linear models as our routers. Thus, $\boldsymbol{\phi}_i \boldsymbol{x} \in \mathbb{R}^{n_i}$ produces gating values which are used to combine the $n_i$ experts at each layer, as in (1), where $\boldsymbol{x}$ is an input token, $\boldsymbol{y}$ is the corresponding layer output and $E_j$ is the $j$th expert.

$$p_j(\boldsymbol{x}) = \frac{(\exp \boldsymbol{\phi}_i \boldsymbol{x})_j}{\sum_k (\exp \boldsymbol{\phi}_i \boldsymbol{x})_k}, \; \boldsymbol{y} = \sum_{j=1}^{n_i} p_j(\boldsymbol{x}) E_j(\boldsymbol{x}) \quad (1)$$

Our experts are simply LoRA modules, which approximate model updates $\Delta \boldsymbol{W} \in \mathbb{R}^{m \times n}$ with a multiplication of two low-rank matrices $\boldsymbol{A} \in \mathbb{R}^{m \times r}$ and $\boldsymbol{B} \in \mathbb{R}^{r \times n}$ with rank $r \ll m, n$. $\boldsymbol{\theta}^G$ and $\boldsymbol{\theta}^S$ are disjoint sets of LoRA A and B matrices.

### 3.2. A Bi-Level Formulation

Instead of learning routing and expert parameters simultaneously like the conventional way in LLMs (Zoph et al., 2022; Fedus et al., 2022a), we update the two sets of parameters in an alternating fashion. We observe *a natural hierarchy between the experts and the router*: the assignment of tokens to experts depends on the router's outputs, while the experts' parameters are updated based on the assigned tokens. In this way, the experts' development follows the router's decisions,

establishing an inherent leader-follower structure. Following Von Stackelberg (2010), we formulate the hierarchical problem as a bi-level optimization objective as follows:

$$\min_{\mathbf{\Phi}} \sum_i \mathcal{L}(\mathbf{X}_i^{\text{valid}}, \mathbf{\Theta}^\star(\mathbf{\Phi}), \boldsymbol{\phi}_i) \qquad \text{(upper)}$$

$$s.t. \ \mathbf{\Theta}^\star(\mathbf{\Phi}) \in \arg\min_{\mathbf{\Theta}} \sum_i \mathcal{L}(\mathbf{X}_i^{\text{train}}, \boldsymbol{\theta}^G, \boldsymbol{\theta}_i^S, \boldsymbol{\phi}_i) \ \text{(lower)}$$

where $\mathcal{L}$ is the language modeling loss. The routing parameters $\mathbf{\Phi} = \{\boldsymbol{\phi}_i\}$ are updated based on the validation loss, which reflects the target distribution (upper optimization), while the expert parameters $\mathbf{\Theta} = \boldsymbol{\theta}^G \cup \{\boldsymbol{\theta}_i^S\}$ are updated using the training loss (lower optimization). This formulation further brings in the following benefits: 1) routing parameters are smaller in size, making them easier to overfit. By separating the two losses, the routing parameters can be updated less frequently using the smaller validation set (visual evidence of less frequent router update leading to improved performance is provided in Figure 8 in the Appendix); 2) this approach handles situations where target distributions differ from training distributions more effectively, as the router outputs (i.e., how the experts should be weighted) can be tailored to specific tasks.

### 3.3. Our Algorithm

To solve our bi-level problem, we use alternating updates of the two sets of parameters. The pseudo-code of our proposed algorithm is detailed in Alg.1 in the Appendix.

**Alternating Update of $\mathbf{\Theta}$ and $\mathbf{\Phi}$.** Alternating update of two sets of parameters is a standard way to solve bi-level optimization problems (Chen et al., 2021). The alternating updates of expert and routing parameters are performed using local training and validation sets separately. To simplify notations, we denote $f_{\text{valid}}(\mathbf{\Theta}, \mathbf{\Phi}) := \sum_i \mathcal{L}(\mathbf{X}_i^{\text{valid}}, \boldsymbol{\theta}^G, \boldsymbol{\theta}_i^S, \boldsymbol{\phi}_i)$ and $f_{\text{train}}(\mathbf{\Theta}, \mathbf{\Phi}) := \sum_i \mathcal{L}(\mathbf{X}_i^{\text{train}}, \boldsymbol{\theta}^G, \boldsymbol{\theta}_i^S, \boldsymbol{\phi}_i)$. Note that in contrast to (upper) bi-level formulation, we allow parameter $\mathbf{\Theta}$ to be free in $f_{\text{valid}}$, which makes it easier to optimize. We can write the alternating update steps as follows.

$$\mathbf{\Phi}_{k+1} = \arg\min_{\mathbf{\Phi}} f_{\text{valid}}(\mathbf{\Theta}_k, \mathbf{\Phi}), \qquad (2)$$

$$\mathbf{\Theta}_{k+1} = \arg\min_{\mathbf{\Theta}} f_{\text{train}}(\mathbf{\Theta}, \mathbf{\Phi}_{k+1}). \qquad (3)$$

Since the updates of $\mathbf{\Theta}$ and $\mathbf{\Phi}$ are disentangled, they do not need to be updated at the same frequency. The routing parameters are smaller in size and thus can be updated less frequently. When updating model parameters, we include an additional load-balancing term as in Fedus et al. (2022a), which is standard in MoE implementation and encourages even distribution of token assignments to experts.

A discussion over the load balancing term is included in Appendix C.4. It is observed that a load-balancing term can improve test performance compared to not having one. However, directing more tokens to the generalists has no noticeable effect.

Given that the data is distributed among clients, when solving optimization problem from equation (3), we first obtain the solutions $\boldsymbol{\theta}_i^G$ and $\boldsymbol{\theta}_i^S$ to local problems, for each client $i$. A parameter aggregation is then performed on the user-specific $\boldsymbol{\theta}_i^G$ via a trusted server to establish a shared $\boldsymbol{\theta}_G$ across all users.

$$\left\{ \tilde{\boldsymbol{\theta}}_i^{G,k+1}, \tilde{\boldsymbol{\theta}}_i^{S,k+1} \right\}_{i=1}^N = \arg\min_{\mathbf{\Theta}} f_{\text{train}}(\mathbf{\Theta}, \mathbf{\Phi}_{k+1}),$$

$$\mathbf{\Theta}^{k+1} = \left( \frac{1}{N} \sum_i \tilde{\boldsymbol{\theta}}_i^{G,k+1}, \{\tilde{\boldsymbol{\theta}}_i^{S,k+1}\} \right). \qquad (4)$$

In the next round, each user replaces their $\boldsymbol{\theta}_i^G$ with the global $\boldsymbol{\theta}_G$, while their $\boldsymbol{\theta}_i^S$ remains local.

### 3.4. Convergence Results

We study the convergence properties of our alternating minimization process. First, we establish a linear rate of convergence under general assumptions on our objectives, that always hold *locally*, when the parameters are close to the training solution (assuming the pretrained model is not far from the fine-tuned models). Then, we show that in the case of *linear experts*, the same optimization procedure possesses *global* linear convergence.

We denote partial minimization operators from (2), (3) by

$$u_1(\mathbf{\Theta}) := \arg\min_{\mathbf{\Phi}} f_{\text{valid}}(\mathbf{\Theta}, \mathbf{\Phi}),$$

$$u_2(\mathbf{\Phi}) := \arg\min_{\mathbf{\Theta}} f_{\text{train}}(\mathbf{\Theta}, \mathbf{\Phi}),$$

and their compositions by $T := u_2 \circ u_1$ and $P := u_1 \circ u_2$. Note that both $T$ and $P$ act on the corresponding spaces of $\mathbf{\Theta}$ and $\mathbf{\Phi}$: $T : \mathbb{R}^{|\mathbf{\Theta}|} \to \mathbb{R}^{|\mathbf{\Theta}|}$ and $P : \mathbb{R}^{|\mathbf{\Phi}|} \to \mathbb{R}^{|\mathbf{\Phi}|}$.

**Assumption 1** (Shared Optima). *There exist $\mathbf{\Theta}^\star$ and $\mathbf{\Phi}^\star$ such that*

$$\mathbf{\Theta}^\star = T(\mathbf{\Theta}^\star) \qquad and \qquad \mathbf{\Phi}^\star = P(\mathbf{\Phi}^\star). \quad (5)$$

Eq. (5) means that $f_{\text{valid}}$ and $f_{\text{train}}$ share the same global optima, which is reasonable when the train and validation data are similar, $\mathbf{X}_i^{\text{train}} \sim \mathbf{X}_i^{\text{valid}}$, and, hence, $f_{\text{valid}} \approx f_{\text{train}}$. It also holds for overparametrized models, such as LLMs.

**Assumption 2** (Contraction Property). *Let $u_1$ and $u_2$ be Lipschitz with some constants $\lambda_1, \lambda_2 > 0$, for any $\mathbf{\Theta}, \bar{\mathbf{\Theta}}$ and $\mathbf{\Phi}, \bar{\mathbf{\Phi}}$:*

$$\|u_1(\mathbf{\Theta}) - u_1(\bar{\mathbf{\Theta}})\| \leq \lambda_1 \|\mathbf{\Theta} - \bar{\mathbf{\Theta}}\|,$$

$$\|u_2(\mathbf{\Phi}) - u_2(\bar{\mathbf{\Phi}})\| \leq \lambda_2 \|\mathbf{\Phi} - \bar{\mathbf{\Phi}}\|. \qquad (6)$$

**Theorem 3.1** (Convergence under Contraction). *If Assumptions 1, 2 hold, and $\lambda_1 \cdot \lambda_2 < 1$, then the weights $(\boldsymbol{\Theta}_k, \boldsymbol{\Phi}_k)$ generated by alternating updates (2), (3) converge to $(\boldsymbol{\Theta}^\star, \boldsymbol{\Phi}^\star)$ with a linear rate.*

The proof is provided in Appendix F.2. We show that the contraction property holds when the objectives are convex quadratics. As a consequence, it also holds locally for any sufficiently smooth models, using their Taylor expansions around a local minimum. Large models are usually smooth, and when initialized with a well-pre-trained model, it suffices to guarantee local convergence.

Our alternating minimization process can guarantee global convergence as well, when the experts are linear models, see Appendix F.4. This indicates a wide applicability of solving MoE objectives via alternating minimization.

**Theorem 3.2** (Global Convergence for Linear Experts). *If $f_{valid} = f_{train}$ and all the expert modules are linear models, we have a global linear convergence rate for a practical instance of our method.*

# 4. Experiments

## 4.1. Setup

### 4.1.1. DATASETS

We selected the following datasets to demonstrate the efficacy of our proposed algorithm: 1) *Multilingual Wikipedia*: Wikipedia articles in four languages: German, French and Italian from Wikimedia-Foundation, and Dutch from Guo et al. (2020); 2) *SlimPajama*: We pick the following four categories – StackExchange, Github Codes, ArXiv, Book from Soboleva et al. (2023); 3) *AG News*: News from categories of World, Sports, Business, and Sci/Tech (Zhang et al., 2016). 4) *Common Corpus* (pleias, 2024): specifically the following three categories – YouTube-Commons, Public Domain Books, and EU Tenders collections, and the Harvard US Patent dataset from Suzgun et al. (2022).

A distinct category is assigned to a user, as it simulates the most challenging scenario for collaboration. Given our emphasis on next token prediction, we anticipate shared predictions among users while maintaining category-specific distinctions. We further create the following two scenarios to showcase the wide applicability of our method:

**In-Distribution Tasks.** For each user, we construct validation and test datasets that follow the same distribution as the training data. We address two scenarios in this context: (i) variation in language usage across users (Multilingual Wikipedia), and (ii) variation in topic coverage across users (SlimPajama, Common-Corpus).

**Out-of-Distribution Tasks.** For each user, we create validation and test datasets from a distribution different from the training data. During training, each user is assigned a

single category, but their validation and test sets consist of a uniform mixture of all categories. This approach accounts for potential shifts in topics within users.

### 4.1.2. EXPERIMENTAL DETAILS

We choose the following base model architectures: GPT2 (124M, English only) and Llama 3.2(1B, Multilingual)[2]. We incorporate LoRA modules into every linear layer, including MLP and Self-Attention Layers, following the recommendations of Fomenko et al. (2024). A routing mechanism is exclusively implemented atop MLP layers. The number of LoRA experts in MLP blocks depends on the local resource abundance. For more experimental details, we refer readers to Appendix B.

## 4.2. Data-Driven Selection: Generalist vs. Specialist

We start by equipping users with the same model architecture locally, to illustrate the effectiveness of our hierarchical learning of routing and expert parameters. We compare our one generalist one specialist (`CoMiGS-1G1S`) method to the following baselines. In order to match the trainable parameter count of our method, we use 2 times LoRA modules within each user.

- *Upper and lower bounds*: 1) `Pretrained`: Pretrained checkpoints. 2) `Centralized`: A single model trained using data from all users. (Note this method is an unrealistic baseline as data cannot leave the devices due to privacy concerns.)

- *Baselines*: 1) `Local`: Training individually using only local data. 2) `FedAvg`: Aggregating LoRA parameters across users using uniform weights, which is equivalent to applying FedAvg (McMahan et al., 2017). 3) `PCL`: Aggregating LoRA parameters using a client-level collaboration graph. The graph is updated using validation performances. (Strategy 2 in Wagner et al. (2024)). 4) `pFedMoE`: We directly apply the method from Yi et al. (2024) in the language domain where we update routing and expert parameters at the same time and choose tokens as a routing unit. 5) `FDLoRA`: Global and local parameters are learned, and a client-wise fusion weight is learned to combine global and local parameters. (Qi et al., 2024)

- *Ablations*: 1) `CoMiGS-2S`: Both experts are specialists, meaning their weights are neither shared nor aggregated. The routing parameters are updated using a separate validation set like in `CoMiGS-1G1S`. 2) `CoMiGS-2G`: Both experts are generalists, meaning their weights are always shared and aggregated. The routing parameters are updated like in `CoMiGS-1G1S`.

---

[2]We adopt the codes from https://github.com/karpathy/nanoGPT and https://github.com/danielgrittner/nanoGPT-LoRA, https://github.com/pjlab-sys4nlp/llama-moe

Table 1: Mean (std) test perplexity over the users with homogeneous models, averaged across 3 seeds (the lower the better). Light grey denotes in-distribution tasks and dark grey denotes out-of-distrition tasks.

| Base Model | GPT2-124M | | | LLAMA3.2-1B | |
|---|---|---|---|---|---|
| Dataset | Multilingual | SlimPajama | AG News | Com-Corpus | AG News |
| Pretrained | 156.12 | 37.19 | 90.65 | 30.40 | 29.37 |
| Centralized | 55.41 (0.12) | 19.53 (0.14) | 28.19 (0.52) | 17.97 (0.19) | 16.12 (0.05) |
| Local | 54.38 (0.32) | 26.95 (0.14) | 41.46 (0.06) | 20.19 (0.11) | 19.96 (0.01) |
| FedAvg | 58.80 (0.34) | 23.27 (0.05) | 31.84 (0.02) | 21.95 (0.11) | 15.86 (0.05) |
| PCL | 54.53 (0.19) | 26.99 (0.19) | 32.25 (0.12) | 19.65 (0.03) | 16.84 (0.05) |
| pFedMoE | 52.27 (0.17) | 25.40 (0.09) | 38.72 (0.21) | 20.41 (0.05) | 17.84 (0.05) |
| FDLoRA | 57.45 (0.81) | 22.71 (0.40) | 33.61 (0.07) | 22.11 (0.05) | 16.64 (0.02) |
| CoMiGS - 2S | **46.36 (0.16)** | 22.51 (0.08) | 35.81 (0.13) | 18.46 (0.13) | 18.03 (0.11) |
| CoMiGS - 2G | 58.31 (0.17) | **21.36 (0.01)** | **31.18 (0.05)** | 20.18 (0.09) | **15.41 (0.05)** |
| CoMiGS - 1G1S | 47.19 (0.10) | 21.79 (0.04) | 33.53 (0.03) | **18.37 (0.03)** | 16.31 (0.05) |

### 4.2.1. RESULT ANALYSIS

The comparison between our method and the baseline methods is summarized in Table 1.

**Effectiveness of Our Routing Mechanism.** Depending on the dataset, either CoMiGS-2G or CoMiGS-2S achieves the highest performance. The key distinction compared to Local or FedAvg is the existence of a layer-wise token-level router, which learns to combine the two generalists or specialists. This emphasizes while knowledge might be present, the way it's combined is the key. Moreover, pFedMoE, despite having a learned router as well, underperforms our method, even in the in-distribution scenario. The reason is that the routing parameters are updated simultaneously with the expert parameters using the training set, and thus cannot effectively adapt to the target distribution. When a separate validation set is not available, CoMiGS can alternatively sample a new training batch to update the routers and still offer competitive results for in-distribution tasks (see Table 5).

**Token-level Collaborative Decisions Outperform Client-Level.** Compared to the state-of-the-art baseline PCL and FDLoRA, our method demonstrates a clear performance improvement. While both methods require a separate validation set as in our method to determine collaboration weights, PCL determines the weights to combine each client's models iteratively while FDLoRA determines the weights for the global and local model at the end of training. Our method, in contrast, decides the collaboration pattern based on each input token, allowing the router weights to co-adapt with the expert parameters throughout training. This enables a more flexible and fine-grained collaboration.

**The Necessity of the Co-existence of Generalists and Specialists.** The performances of CoMiGS-2G and CoMiGS-2S are not consistent across the different scenarios, while our CoMiGS-1G1S can always closely track the best-performing model, which is clearly visualized in Figure 9. Even for in-distribution tasks, it is unclear whether CoMiGS-2G or CoMiGS-2S will outperform, suggesting both generalists and specialists are necessary as it is im-

possible to determine the language structure in advance. Even drastically different users still share many of the same tokens. A data-dependent combination of generalists and specialists is required.

### 4.2.2. ROUTING ANALYSIS

**Token-wise Analysis.** Figure 3 visualizes token-level routing results for models fine-tuned on the SlimPajama dataset. In the *first* layer, function words (e.g., "and," "a," "on," "the") are predominantly routed to generalists. In contrast, in the *last* layer, content words are more frequently assigned to generalists. This pattern is particularly evident for the first two users, trained on math and programming texts, where domain-specific terms are primarily routed to specialists. These findings suggest that experts in later layers develop more distinct role specializations. Importantly, only the top choice is highlighted here. The abundance of blue does not imply that generalist experts play no role in predicting the next token. As compared to when only specialists are present (CoMiGS-2S), our CoMiGS-1G1S gives more consistent results. More detailed token-wise routing result visualization including out-of-distribution tasks can be seen in Appendix E. When dealing with out-of-distribution texts, there is an increasing tendency to seek generalists, as shown in the off-diagonal entries in Figure 14-19.

**Layer-wise Analysis.** Figure 4 depicts the evolution of averaged layer-wise router outputs for the generalist and specialist experts on the *out-of-distribution* task, comparing CoMiGS-1G1S and pFedMoE. As training progresses, CoMiGS-1G1S undergoes a *phase transition*: the layer-wise routers initially favor generalists but gradually shift towards specialists. This shift is not observed in pFedMoE, highlighting the critical role of our routing mechanism in handling out-of-distribution tasks. Additionally, we notice different layers converge to a different expert score distribution. When applying our CoMiGS-1G1S, for each user, there are always certain layers where the routers consistently prefer generalists, which aligns with the fact that our target distribution is a union of all local training distributions. For *in-distribution* tasks (Figure 10), during early stage of training, some layers favor generalists. When close to convergence, all layers favor specialists. We attribute this to the fact that generalists are updated with more tokens and are thus knowledgable from an early stage, while it takes longer for specialists to refine their knowledge with small local training data.

### 4.3. Adaptation to Computational Resource Heterogeneity

#### 4.3.1. BASELINE COMPARISON

In this section, our focus is to deal with computational resource heterogeneity, where users can have different num-

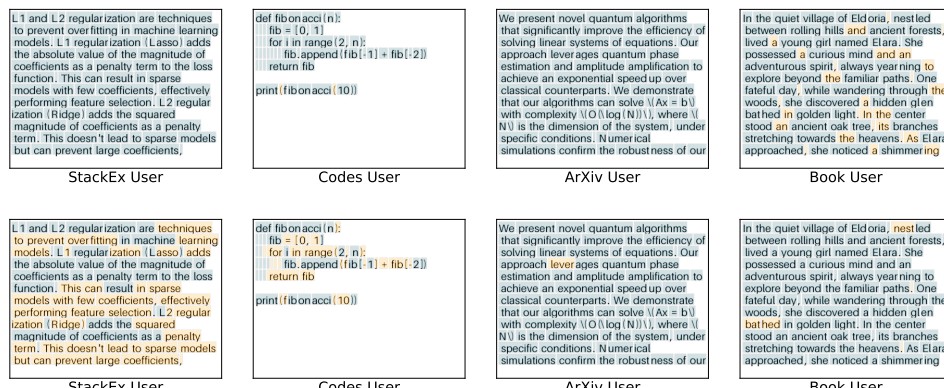

Figure 3: Visualization of in-distribution token-level routing results for `CoMiGS-1G1S` trained on SlimPajama. Tokens are colored with the Top1 expert choice at the first layer (top) and last layer (bottom). Orange denotes the generalist and blue denotes the specialist. Texts are generated by ChatGPT. Further colored text plots are provided in Appendix E.

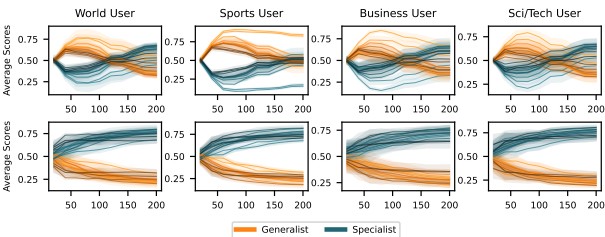

Figure 4: Expert Scores for the *generalist* expert and the *specialist* expert, averaged across all tokens and multiple batches for the out-of-distribution task (AG News). X-axis: number of iterations. Top: `CoMiGS-1G1S`, Bottom: `pFedMoE`. Darker colors indicate deeper layers.

bers of experts $n_i$. We denote different experimental setup by specifying the list of $n_i$s. We still keep one generalist expert per device, but the number of specialists can vary across the users (the variation is called One-Generalist-X-Specialists, in short, `CoMiGS-1GXS`). It's important to note that the richness of computational resources doesn't always correlate with the complexity of local data. For instance, some users may have ample computational resources but local data in small quantities. In such cases, a crucial objective is to prevent overfitting due to redundant model-fitting abilities.

We compare our approach to two state-of-the-art baselines: `HetLoRA` from Cho et al. (2023) and `FlexLoRA` from Bai et al. (2024), both of which adapt LoRA ranks based on the resource capacity of each user. `HetLoRA` aggregates LoRA matrices $A$ and $B$ by zero-padding to the maximum rank and then distributes them back using rank truncation. In contrast, `FlexLoRA` first reconstructs model updates $\Delta W$ and redistributes the aggregated updates using SVD. We compare our method to these baselines by matching the number of tunable parameters, measured as both active and full parameters. For example, to match the full parameter

count of `CoMiGS-1GXS` with $(4, 2, 2, 2)$ LoRA experts (rank 8), LoRA modules of ranks $(32, 16, 16, 16)$ would be required. With Top2 routing, to match the active parameter count, each user would need LoRA modules of rank 16.

Our results, presented in Table 2, are based on allocating varying numbers of experts to users, with computational resource availability decoupled from local task complexity. Our method outperforms the baseline methods for all *in-distribution* tasks, regardless of matching the full parameter count or the active parameter count. This advantage stems from the fact that both `HetLoRA` and `FlexLoRA` average model parameters across users without allocating parameters for local adaptations, focusing on building a strong generalist model. In contrast, our approach adaptively integrates both generalist and specialist knowledge, excelling in scenarios where specialized knowledge is crucial.

Table 2: Mean test ppl (std) over users with heterogeneous models, averaged across 3 seeds. Light / dark grey denote in-distribution and out-of-distribution tasks respectively.

| | OURS | HETLORA | | FLEXLORA | |
| | COMIGS-1GXS | ACTIVE | FULL | ACTIVE | FULL |
|---|---|---|---|---|---|
| *GPT2-124M* | | | | | |
| **MULTILINGUAL** | | | | | |
| *(2,2,4,4)* | **46.48 (0.16)** | 57.76 (0.10) | 58.60 (0.20) | 77.71 (0.15) | 77.66 (0.06) |
| *(4,4,2,2)* | **47.24 (0.09)** | 57.76 (0.10) | 59.14 (0.04) | 77.71 (0.15) | 75.64 (0.19) |
| **SLIMPAJAMA** | | | | | |
| *(2,4,4,2)* | **22.10 (0.17)** | 23.33 (0.10) | 23.15 (0.09) | 22.98 (0.10) | 23.03 (0.07)) |
| *(4,2,2,4)* | **22.28 (0.09)** | 23.33 (0.10) | 23.17 (0.09) | 22.98 (0.10) | 23.03 (0.08) |
| **AG NEWS** | | | | | |
| *(4,2,2,2)* | 33.66 (0.07) | **31.58 (0.14)** | 31.95 (0.13) | 36.41 (0.18) | 36.62 (0.11) |
| *(2,4,4,4)* | 34.22 (0.09) | **31.58 (0.14)** | 32.52 (0.19) | 36.41 (0.18) | 36.46 (0.04) |
| *Llama3.2-1B* | | | | | |
| **COMMON-CORPUS** | | | | | |
| *(2,4,4,2)* | **18.74 (0.14)** | 21.41 (0.12) | 21.74 (0.09) | 24.63 (0.12) | 25.18 (0.08) |
| *(4,2,2,4)* | **18.68 (0.11)** | 21.41 (0.12) | 21.61 (0.10) | 24.63 (0.12) | 24.74 (0.09) |
| **AG NEWS** | | | | | |
| *(4,2,2,2)* | 16.39 (0.11) | **15.89 (0.05)** | 16.02 (0.05) | 17.33 (0.04) | 17.52 (0.04) |
| *(2,4,4,4)* | 16.44 (0.07) | **15.89 (0.05)** | 16.25 (0.11) | 17.33 (0.04) | 17.70 (0.10) |

## 4.4. User-specific Analysis

In this section, we investigate how each user can benefit from our `CoMiGS-1GXS`. It is observed that our approach is more robust to overfitting due to the regularizing effect

of the generalist, while at the same time better fitting local data through the incorporation of specialist knowledge.

We conduct experiments using the Multilingual Wikipedia dataset, where there are enough tokens to allocate different data quantities to users. In practice, users may not know their local data complexity, leading to a potential mismatch in resource allocation relative to data quantity. To simulate such scenarios, we allocate model capabilities—measured by $n_i$ (the number of LoRA modules per user)—either positively or negatively correlated with their local data size. It is important to note that one generalist is always assigned. Top2 routing is always performed is $n_i \geq 2$.

**More Specialists Help with Higher Data Quantity.** High data quantity users (French and Italian) consistently benefit from having more specialists locally, as their test perplexities decrease when the number of specialists increases from 1 to 3 to 7. This suggests that when sufficient local training data is available, adding more specialists leads to improved performance.

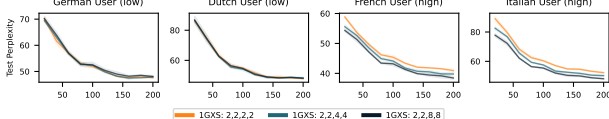

Figure 5: Test Perplexity vs. the number of iterations. Low and high denote data quantity. Legend denotes $n_i$.

**Generalists Help to Prevent Redundant Specialists from Over-Fitting.** For users with low data quantities, local model training with just two LoRA modules already results in overfitting (a trend observed in Figure 9). Our method succeeds to suppress overfitting, even when fine-tuning twice or four times as many expert parameters. We attribute this to the existence of the generalists.

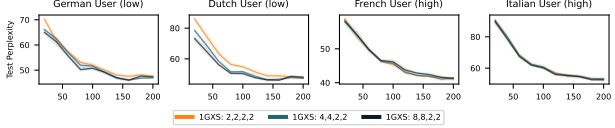

Figure 6: Test Perplexity vs. the number of iterations. Low and high denote data quantity. Legend denotes $n_i$.

**Specialists Can Benefit Generalists.** What happens if users can only support a maximum of one expert? In our setup, such users must rely on the generalist expert when participating in collaboration. Interestingly, even when their collaborators are allocated more specialists, low-resourced users with only one generalist still benefit from the refined role diversification between generalists and specialists. As a result, the generalists become more powerful, as demonstrated in Figure 7.

We provide an additional example of the impact of local data quantities in Appendix D using SlimPajama dataset.

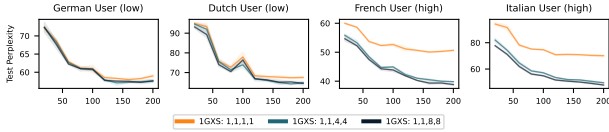

Figure 7: Test Perplexity vs. the number of iterations. Low and high denote data quantity. Legend denotes $n_i$.

Similar conclusions can be drawn from our empirical results. However, there is a limit to how much generalists can help prevent overfitting when the local tasks are easy.

### 4.5. Computational and Communication Overhead

Our approach offers a significant advantage for on-device deployment due to its minimal computational and communication overhead. We compare the resource consumption of our `CoMiGS-1G1S` to `FedAvg` in Table 3, matching the parameter count for LoRA modules.

The communication costs are halved compared to standard `FedAvg`, as only the weights of generalist experts are exchanged. Our framework employs a first-order algorithm, ensuring that computation and memory requirements remain on par with those of standard `FedAvg` algorithms. The additional memory and computational overhead primarily stem from the inclusion of the router, which is minimal (1.25% increase) since the router consists of a single-layer MLP.

Table 3: Extra resource consumption (per device) `CoMiGS-1G1S` compared to standard `FedAvg`, assuming base model is GPT-124M with bfloat16 training.

| COMP. OVERHEAD / FORWARD PASS | MEMORY | COMM. COSTS / ROUND |
|---|---|---|
| + 5 MFLOPS (+1.25%) | + 0.035 MB (+1.25%) | -1.41 MB (-50%) |

## 5. Conclusions

We propose a novel framework for on-device personalized collaborative fine-tuning of LLMs, grounded in an innovative bi-level formulation of the Mixture-of-Experts learning objective. Our fine-grained integration of generalist and specialist expert knowledge achieves superior performance in balancing personalization and collaboration within Federated LLMs.

Furthermore, our framework is the first to address both model and data heterogeneity in collaborative LLM training. It further decouples local data quantity from resource availability, allowing high-resourced users to leverage larger datasets for improved performance while remaining resilient against overfitting in low-data scenarios. `CoMiGS` is both theoretically sound and resource-efficient for practical deployment.

## Impact Statement

We offer a collaboration framework for edge devices, aiming to enable smaller devices to leverage large language models (LLMs) despite limited resources and data availability. Our approach enhances fairness and mitigates privacy concerns by ensuring data remains on end devices. The privacy aspects can further be enhanced by differential private aggregation of generalist weights, which we do not pursue here.

The robustness towards attackers is beyond the scope of our work. Our collaboration framework has no guarantee of resilience towards adversarial attackers through the aggregation of the generalist weights, which could potentially lead to misuse by certain parties. Further research is required on top of our framework to guarantee its safe deployment.

## Acknowledgements

This work was supported by the Swiss State Secretariat for Education, Research and Innovation (SERI) under contract number 22.00133 and received funding from the European Union's Horizon 2020 research and innovation programme under grant agreement No 101017915 (DIGIPREDICT).

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

# A. Our Algorithm

The pseudo codes of our proposed CoMiGS method are presented in Alg. 1. While the scheme requires a server, it can alternatively be implemented in a serverless all2all fashion, which requires $N$ times more communication overhead and we do not further pursue this here.

---

**Algorithm 1** Pseudo code of our proposed algorithm

---

**Input:** Expert parameters $\{\boldsymbol{\theta}_{i,0}^G, \boldsymbol{\theta}_{i,0}^S\}$, routing parameters $\{\boldsymbol{\phi}_{i,0}\}$. Local training data and validation data $\{\boldsymbol{X}_i^{\text{train}}, \boldsymbol{X}_i^{\text{valid}}\}$, $i \in \{1, 2, .., N\}$. Communication round $T$ and routing update period $\tau$. Load balancing weight $\lambda$.
**for** $t = 1, ..., T$ **do**
    Server aggregates generalist parameters: $\boldsymbol{\theta}_{t-1}^G = \frac{1}{N} \sum_i \boldsymbol{\theta}_{i,t-1}^G$
    **for** $i \in [0, N)$ **do**
        Users download aggregated generalist weights and
        prepare model parameters for training $\{\boldsymbol{\theta}_{t-1}^G, \boldsymbol{\theta}_{i,t-1}^S, \boldsymbol{\phi}_{i,t-1}\}$
        Do gradient steps on $(\boldsymbol{\theta}_{t-1}^G, \boldsymbol{\theta}_{i,t-1}^S)$ towards minimizing (7) and get $(\boldsymbol{\theta}_{i,t}^G, \boldsymbol{\theta}_{i,t}^S)$

$$
\begin{aligned}
\min_{\boldsymbol{\theta}_i^G, \boldsymbol{\theta}_i^S} & \mathcal{L}(f(\boldsymbol{X}_i^{\text{train}}; \boldsymbol{\theta}_i^G, \boldsymbol{\theta}_i^S, \boldsymbol{\phi}_{i,t-1}), \boldsymbol{X}_i^{\text{train}}) + \\
& \lambda \cdot \mathcal{L}_i^{\text{LB}}(\boldsymbol{X}_i^{\text{train}}; \boldsymbol{\theta}_i^G, \boldsymbol{\theta}_i^S, \boldsymbol{\phi}_{i,t-1})
\end{aligned}
\tag{7}
$$

        **if** $t \% \tau = 0$ **then**
            Do gradient steps on $\boldsymbol{\phi}_{i,t-1}$ towards minimizing (8) and get $\boldsymbol{\phi}_{i,t}$

$$
\begin{aligned}
\min_{\boldsymbol{\phi}_i} & \mathcal{L}(f(\boldsymbol{X}_i^{\text{valid}}; \boldsymbol{\theta}_{i,t}^G, \boldsymbol{\theta}_{i,t}^S, \boldsymbol{\phi}_i), \boldsymbol{X}_i^{\text{valid}}) + \\
& \lambda \cdot \mathcal{L}_i^{\text{LB}}(\boldsymbol{X}_i^{\text{valid}}; \boldsymbol{\theta}_{i,t}^G, \boldsymbol{\theta}_{i,t}^S, \boldsymbol{\phi}_i)
\end{aligned}
\tag{8}
$$

        **end if**
    **end for**
    Each device $i \in \{1, 2, .., N\}$ sends generalist weights $\boldsymbol{\theta}_{i,t}^G$ to the server
**end for**
**Return:** Expert parameters $\{\boldsymbol{\theta}_{i,T}^G, \boldsymbol{\theta}_{i,T}^S\}$ and routing parameters $\{\boldsymbol{\phi}_{i,T}\}$

---

# B. Extra Experimental Details

### B.1. Training Details

Following Kalajdzievski (2023), we choose $\gamma$ to be a rank-stabilized value, a technique which helps stabilize gradient norms. $\alpha$ and the rank $r$ are hyper-parameters to choose from. The LoRA modules function as follows:

$$
\boldsymbol{W} = \boldsymbol{W}^0 + \gamma \cdot \boldsymbol{A}\boldsymbol{B}, \qquad \gamma = \frac{\alpha}{\sqrt{r}}
\tag{9}
$$

All our experiments except the centralized ones were conducted on a single A100-SXM4-40GB GPU. The centralized learning baseline experiments were conducted on a single A100-SXM4-80GB GPU, as a batch size of 64*4 requires a larger storage capacity.

We use a constant learning rate of $2 \times 10^{-3}$ for updating routing parameters and a $2 \times 10^{-3}$ learning rate with a one-cycle cosine schedule for expert parameters during fine-tuning. The LoRA rank $r$ is set to 8 unless otherwise specified, with LoRA alpha $\alpha$ set to 16, following the common practice of setting alpha to twice the rank (Raschka, 2023). A load balancing weight 0.01 is always applied.

**GPT2 Experiments.** For AG News and Multilingual Wikipedia data splits, we conduct 20 communication rounds. For SlimPajama data splits, due to greater category diversity, we conduct 50 communication rounds. Between each pair of communication rounds, there are 10 local iterations. In each iteration, a batch size of 64 is processed with a context length

of 128. We set the routing update period to 30 iterations, and every time we update routing parameters, we do 10 gradient steps on the validation loss. The choice of the hyperparamters is from a sweep run and we provide the evidence in Figure 8.

**Llama3.2 Experiments.** For AG News data splits, we conduct 10 communication rounds. For Common-corpus data splits, due to greater category diversity, we conduct 20 communication rounds. Between each pair of communication rounds, there are 10 local iterations. In each iteration, a batch size of 64 is processed with a context length of 128. We set the routing update period to 30 iterations, and every time we update routing parameters, we do 10 gradient steps on the validation loss.

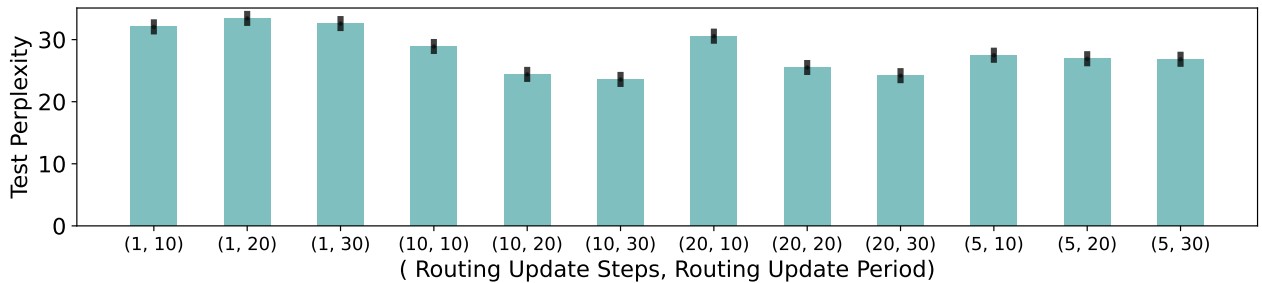

Figure 8: Sweep results on SlimPajama data splits using GPT2-124M base model. We ablate the impact of the update period ($\tau$) and the number of update steps ($s$) on model performance.

### B.2. Datasets

The number of tokens for our experiments within each user is shown in Table 4.

Given the extensive pre-training of Llama 3.2 models on over 15 trillion tokens from public sources, and the multilingual capabilities of Llama 3.2 - 1B, fine-tuning on multilingual Wikipedia or SlimPajama resulted in negligible improvements likely due to significant overlap with the pre-training data corpus. We curated another more difficult fine-tuning dataset – Common Corpus to show case the distinctions of the baseline methods.

Table 4: Number of tokens in each dataset splits

|  |  | User 1 | User 2 | User 3 | User 4 |
|---|---|---|---|---|---|
| **Multilingual** | TRAINING | 557'662 | 407'498 | 556'796 | 451'584 |
|  | VALIDATION | 300'764 | 216'318 | 220'071 | 165'984 |
|  | TEST | 229'720 | 219'741 | 210'570 | 172'547 |
| **SlimPajama** | TRAINING | 1'000'000 | 1'000'000 | 1'000'000 | 1'000'000 |
|  | VALIDATION | 200'000 | 200'000 | 200'000 | 200'000 |
|  | TEST | 200'000 | 200'000 | 200'000 | 200'000 |
| **AG News** | TRAINING | 761'924 | 756'719 | 814'131 | 771'460 |
|  | VALIDATION | 48'809 | 48'730 | 50'398 | 48'249 |
|  | TEST | 48'167 | 47'721 | 48'344 | 49'377 |
| **Common Corpus** | TRAINING | 1'000'000 | 1'000'000 | 1'000'000 | 1'000'000 |
|  | VALIDATION | 200'000 | 200'000 | 200'000 | 200'000 |
|  | TEST | 200'000 | 200'000 | 200'000 | 200'000 |

## C. More Tables and Figures

### C.1. Learning Curves of Different Methods

See Figure 9.

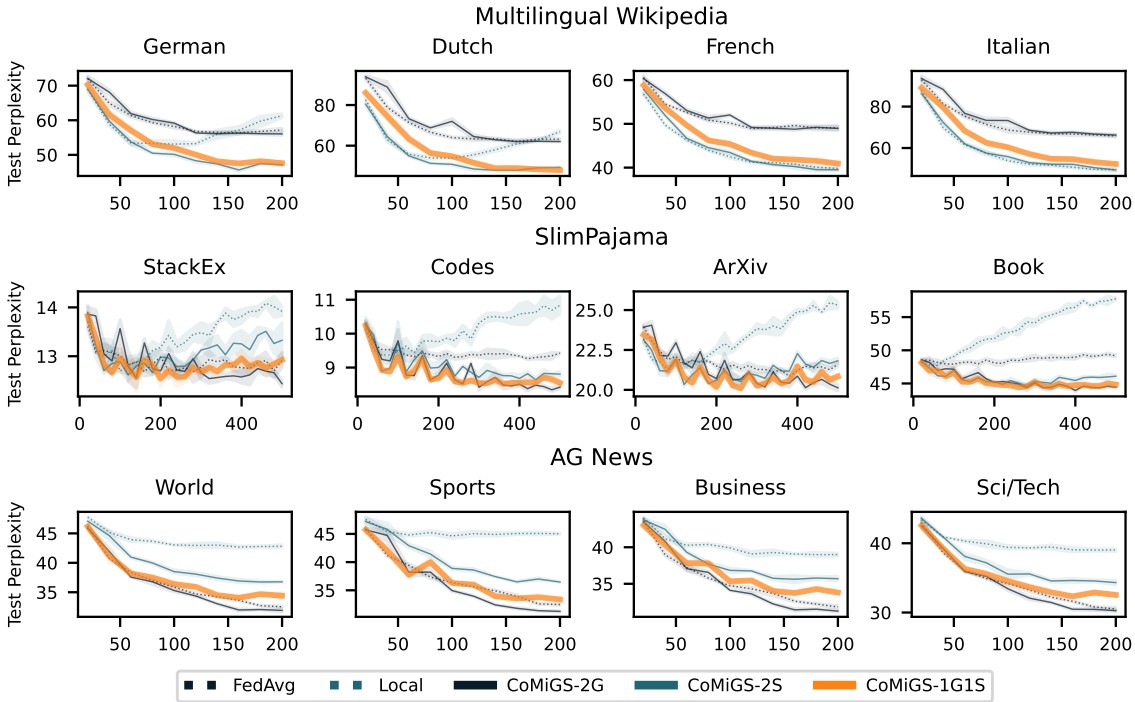

Figure 9: Test Perplexity during training (base model: GPT2-124M): our method closely follows the best performing method

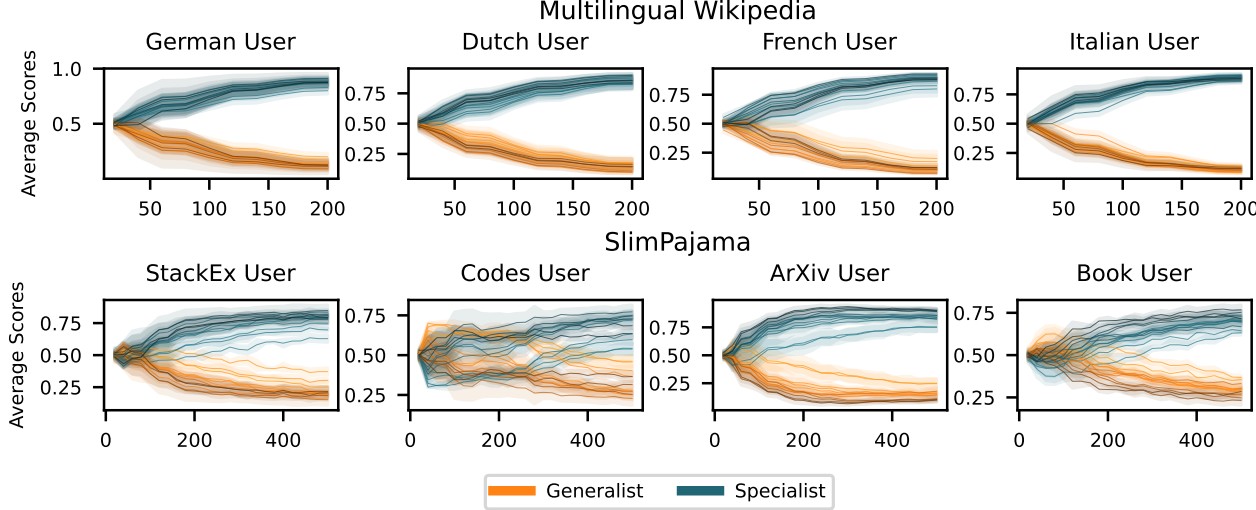

Figure 10: Expert Scores for the *generalist* expert and the *specialist* expert from our `CoMiGS-1G1S` method, averaged across all tokens and multiple batches for the in-distribution task, with x-axis being the number of iterations. Darker colors represent deeper layers.

## C.2. Extended Baseline Comparison

An extended version of Table 1 is presented in Table 5. In this extension, we incorporate two additional ablations: 1) Integration of a routing mechanism, updated simultaneously with the expert networks; 2) Iterative updates alternating between routing and expert parameters, with the routing parameters updated using newly-sampled training batches instead of a dedicated validation set. 2) is to address the scenario where a validation set is not available.

Moreover, we include two other baseline methods – `FFA-LoRA` from Sun et al. (2024) and `FedSA` from Guo et al. (2024). `FFA-LoRA` keeps the LoRA A matrices fixed at initialization, while `FedSA` always aggregates LoRA A matrices but leave LoRA B matrices localized.

Notably, the comparison between scenarios ii) and iii) reveals minimal disparity, underscoring the significance of having an independent validation set exclusively for routing parameter updates.

Table 5: Mean test perplexity over users with homogenous models, averaged across 3 seeds. Mean (std) with a rank locator for the mean (the lower the better). Green denotes the best performing methods and red denotes our method.

| | IN DISTRIBUTION | | OUT OF DISTRIBUTION |
| | *Multilingual* | *SlimPajama* | *AG News* |
|---|---|---|---|
| I) WITHOUT ROUTING | | | |
| *Pretrained* | 156.12 | 37.19 | 90.65 |
| *Centralized* | 55.41 (0.12) | 19.53 (0.14) | 28.19 (0.52) |
| *Local* | 54.38 (0.32) | 26.95 (0.14) | 41.46 (0.06) |
| *FedAvg* | 58.80 (0.34) | 23.27 (0.05) | 31.84 (0.02) |
| *FFA-LoRA* | 66.80 (0.20) | 22.85 (0.12) | 33.13 (0.09) |
| *FedSa-LoRA* | 57.60 (0.14) | 23.40 (0.13) | 31.57 (0.10) |
| *PCL* | 54.53 (0.19) | 26.99 (0.19) | 32.25 (0.12) |
| II) UPDATE ROUTING AND EXPERT PARAMS SIMULTANEOUSLY ON TRAINING LOSS | | | |
| *Local-MoE* | 55.27 (0.40) | 27.16 (0.16) | 41.49 (0.01) |
| *FedAvg-MoE* | 56.77 (0.37) | 23.32 (0.07) | 32.24 (0.08) |
| *pFedMoE* | 52.27 (0.17) | 22.91 (0.18) | 38.72 (0.21) |
| III) ALTERNATING UPDATE ROUTING PARAMS ON NEWLY SAMPLED BATCHES FROM TRAINING SET | | | |
| *Local-MoE - tr* | 53.78 (0.33) | 27.78 (0.06) | 41.46 (0.03) |
| *FedAvg-MoE - tr* | 59.39 (0.13) | 23.00 (0.01) | 31.70 (0.16) |
| *CoMiGS - tr* | 50.86 (0.14) | 25.45 (0.01) | 38.93 (0.08) |
| IV) ALTERNATING UPDATE ROUTING PARAMS ON A VALIDATION SET | | | |
| *CoMiGS - 2S* | 46.36 (0.16) | 22.51 (0.08) | 35.81 (0.13) |
| *CoMiGS - 2G* | 58.31 (0.17) | 21.36 (0.01) | 31.18 (0.05) |
| *CoMiGS - 1G1S* | 47.19 (0.10) | 21.79 (0.04) | 33.53 (0.03) |

## C.3. HetLoRA

Analogously to the baseline experiment comparison in FlexLoRA (Bai et al., 2024), we use $\gamma = 0.99$ as pruning strength and sweep the regularization parameter in $\{5 \times 10^{-2}, 5 \times 10^{-3}, 5 \times 10^{-4}\}$.

## C.4. Is the Standard Load Balancing Loss Sufficient?

The standard load balancing loss encourages equal assignment of tokens to each expert. When the number of experts gets larger, there might not be enough tokens routed to the generalists, which might lead to a under-developed general knowledge. We will verify if this is indeed true.

To encourage enough tokens to be routed to the generalist expert such that more general knowledge can be developed, we modify our load-balancing loss by introducing importance weighting. As we separate the 0-th expert to be the generalist expert and conduct Top-2 routing, the modified load balancing loss is as follows:

$$\mathcal{L}_i^{\text{LB}} = \frac{1}{(n_i - 1)^2 + 1} \cdot f_0 \cdot P_0 + \sum_{j=1}^{n_i - 1} \frac{n_i - 1}{(n_i - 1)^2 + 1} \cdot f_j \cdot P_j \tag{10}$$

Table 6: Test perplexity with different load balancing terms with (hetero) or without (homo) resource heterogeneity.

|  | **No LB** | **LB (uniform)** | **LB (generalist-favored)** |
| --- | --- | --- | --- |
| AG News (homo) | 33.69 (0.21) | 33.53 (0.03) | 33.53 (0.03) |
| AG News (hetero) | 34.31 (0.05) | 34.28 (0.11) | 34.22 (0.09) |
| Multi-Wiki (homo) | 47.31 (0.15) | 47.19 (0.10) | 47.19 (0.10) |
| Multi-Wiki (hetero) | 46.36 (0.16) | 46.15 (0.04) | 46.48 (0.16) |
| SlimPajama (homo) | 21.77 (0.02) | 21.79 (0.04) | 21.79 (0.04) |
| SlimPajama (hetero) | 22.15 (0.07) | 22.10 (0.11) | 22.10 (0.17) |

where

$$f_j = \frac{1}{T} \sum_{x \in \mathcal{B}} \mathbb{1}\{j \in \text{Top2 indices of } p(x)\} \qquad P_j = \frac{1}{T} \sum_{x \in \mathcal{B}} p_j(x) \tag{11}$$

$j$ is the expert index and $p(x) = [p_j(x)]_{j=1}^{n_i}$ is the logit output from the routing network for a specific token $x$. The idea is that one of the top 2 tokens should always be routed to the generalist expert, i.e. the 0-th expert. Thus, $\frac{p_0}{1/2}$ should be equal to $\frac{p_i}{1/2(n_i-1)}$ for $i \neq 0$. As the original load balancing loss encourages uniform distribution, this modification encourages the generalist expert to have a routing probability of 0.5 on expectation. Note that when $n_i = 2$, this $\mathcal{L}_i^{\text{LB}}$ is the same as the original load balancing loss as proposed in Fedus et al. (2022a).

We present the results in Table 6: in both scenarios, whether users have the same or different numbers of experts, including a load-balancing term leads to a slight improvement compared to omitting it. However, encouraging more tokens to be routed to the generalists does not make a significant difference.

## D. Additional Experiments

We replicate the experiments in Section 4.3 with the SlimPajama dataset, where we assign four times as many tokens to ArXiv User and Book User as to Stack Exchange User and Codes User.

**More Specialists Help with Higher Data Quantity.** From Figure 11, it is evident that ArXiv User and Book User, with abundant local data, benefit from having more local experts.

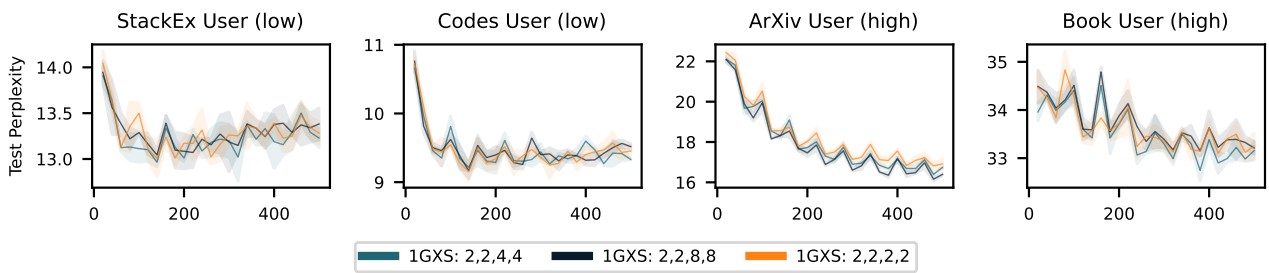

Figure 11: Test Perplexity during training for the SlimPajama setup. ArXiv User and Book User have more local data and thus benefit from having more experts. The numbers in the legend indicate the number of experts $n_i$ within each user. Top-2 routing is performed.

**Generalists Help to Prevent Redundant Specialists from Over-Fitting?** From Figure 12, we observe more prominent overfitting than in Figure 6, likely because the tasks are objectively easier, as indicated by lower test perplexity from the beginning of fine-tuning. Generalists have limited power to prevent overfitting with easy tasks.

**Specialists Can Benefit Generalists.** Low-resourced users that can only support a single expert setup still benefit from collaboration, as the generalist knowledge is refined through a more detailed distinction between specialist and generalist roles via other high-resourced users. This is indicated by the enhanced performances for Stack Exchange and Codes Users.

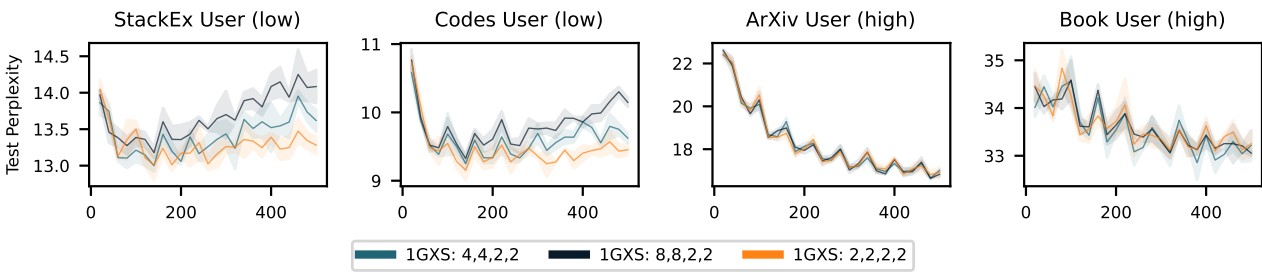

Figure 12: In this SlimPajama setup, Stack Ex User and Codes User despite having low resources locally, overfit slightly on their small-sized local data. Numbers in the legend denote the number of experts $n_i$ within each user. Top2 routing is performed.

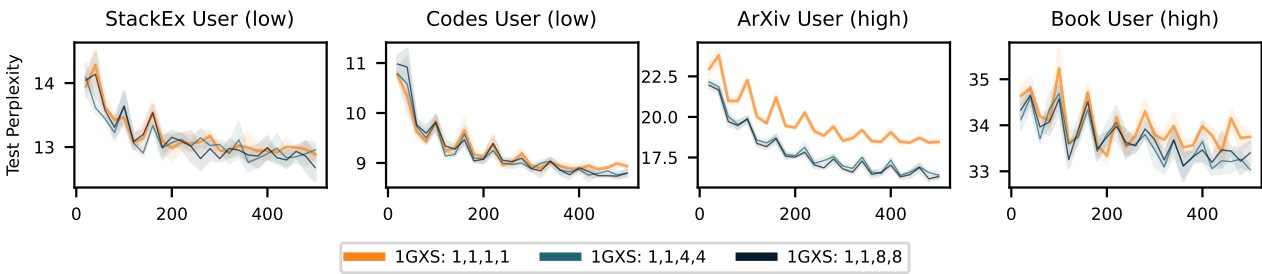

Figure 13: In this SlimPajama setup, Stack Ex User and Codes User, despite having only one expert locally, still benefit from other users having more experts, thereby enhancing the generalist's performance. The numbers in the legend indicate the number of experts, $n_i$, within each user. Top-2 routing is applied when $n_i \geq 2$

## E. Visualization of Expert Specialization

To visualize which tokens are routed to the generalist and specialist experts for our `CoMiGS-1G1S` model trained on SlimPajama, we ask ChatGPT to generate texts in the style of StackExchange, Python Codes, ArXiv Paper and Books. We then feed those texts to the user-specific models and color the token with the Top1 routed index. The routing results after the very first layer (0th), a middle layer (5th), and the very last layer (11th) are presented in Figure 14, 15 and 16.

We perform the same experiments on AG News, asking ChatGPT to generate News text on the topics World, Sports, Business, and Sci/Tech. The routing results after the very first layer (0th), a middle layer (5th), and the very last layer (11th) are presented in Figure 17, 18 and 19.

For all the plots, diagonal entries are *in-distribution* texts and off-diagonal entries are *out-of-distribution* texts.

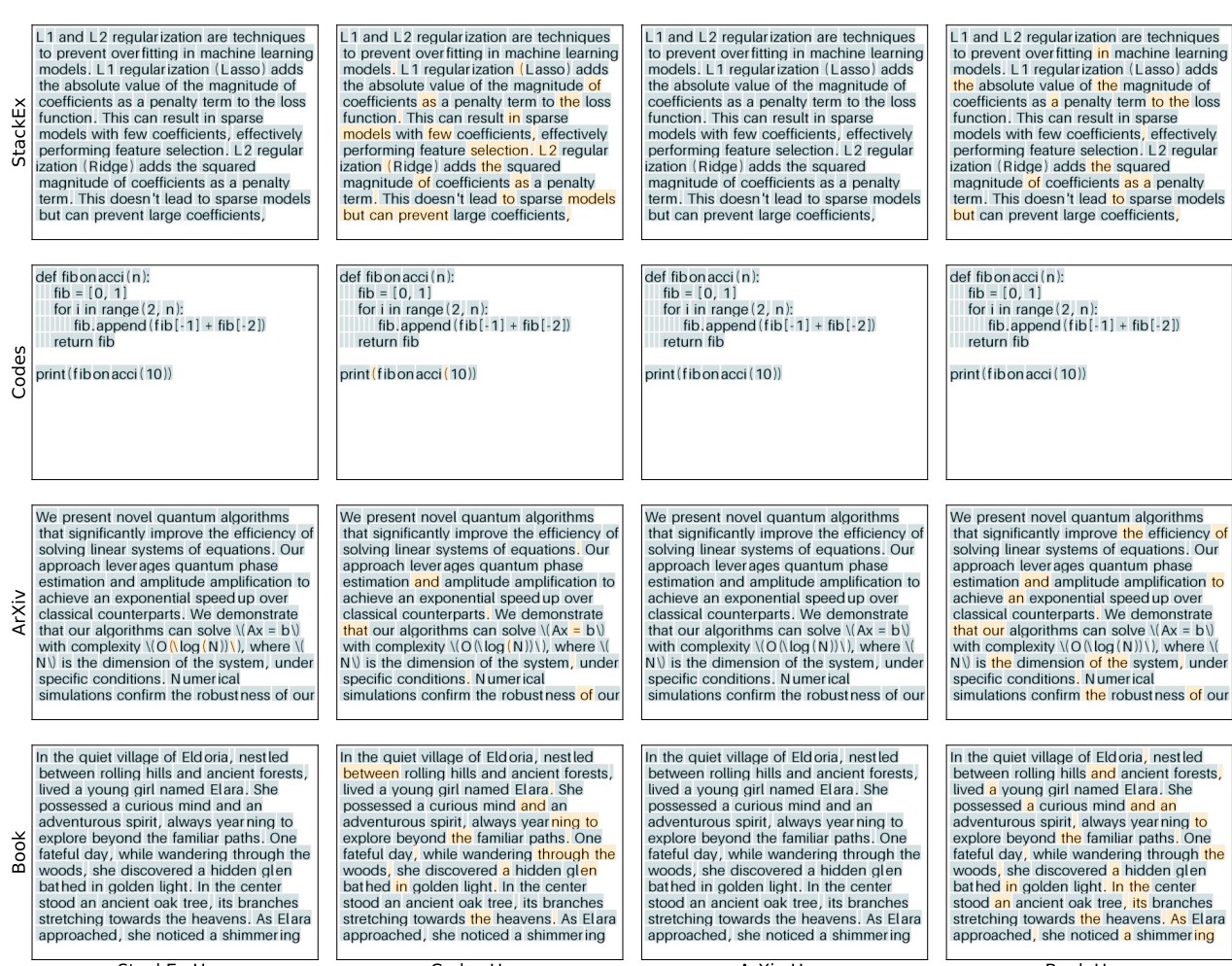

Figure 14: Visualization of token-level routing results for `CoMiGS-1G1S` trained on SlimPajama. Tokens are colored with the first expert choice at the 0th (first) layer. Orange denotes the generalist and blue denotes the specialist. Diagonal entries are in-distribution texts and off-diagonal entries are out-of-distribution texts. Texts are generated by ChatGPT.

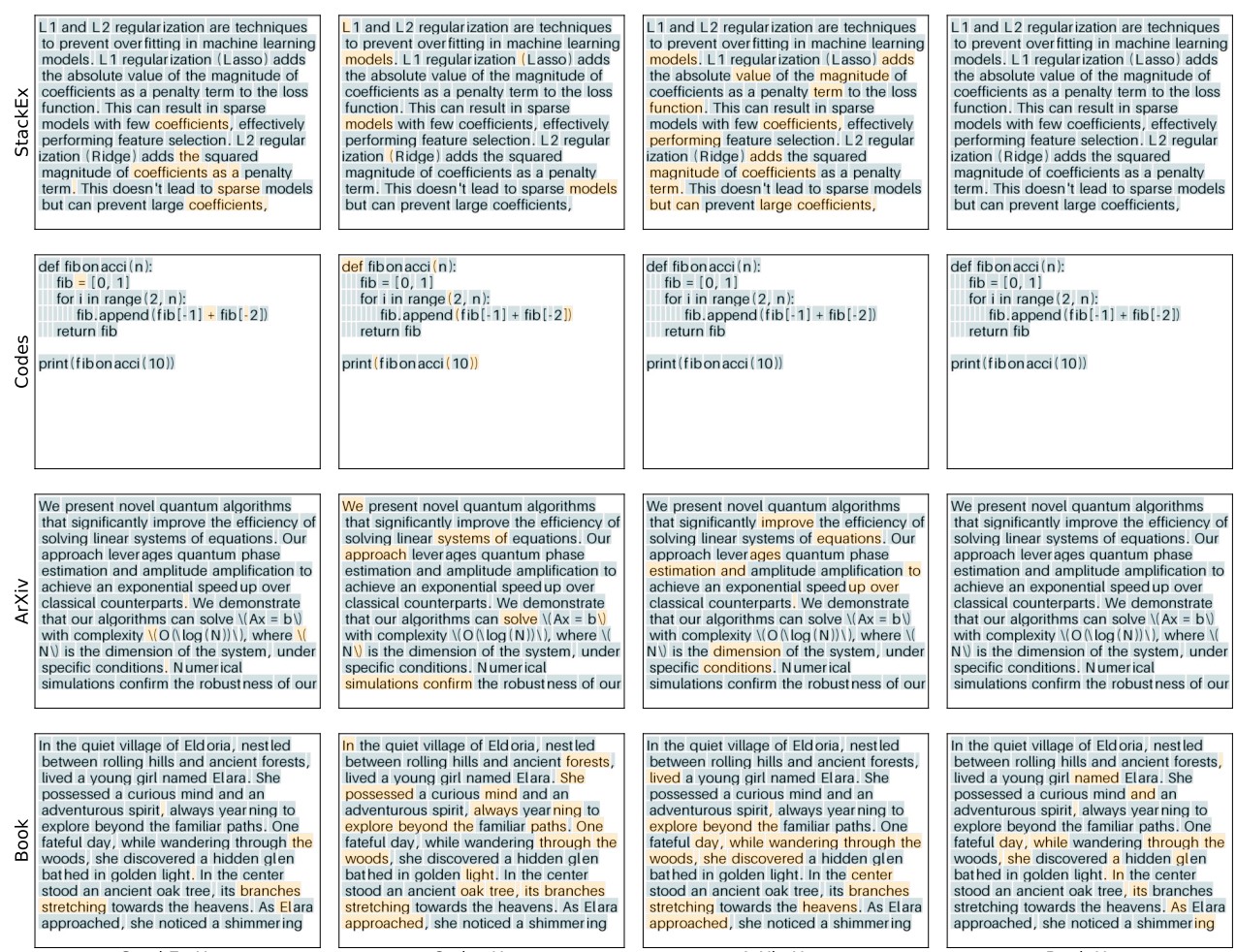

Figure 15: Visualization of token-level routing results for `CoMiGS-1G1S` trained on SlimPajama. Tokens are colored with the first expert choice at the 5th layer. Orange denotes the generalist and blue denotes the specialist. Diagonal entries are in-distribution texts and off-diagonal entries are out-of-distribution texts. Texts are generated by ChatGPT.

Figure 16: Visualization of token-level routing results for `CoMiGS-1G1S` trained on SlimPajama. Tokens are colored with the first expert choice at the 11th (last) layer. Orange denotes the generalist and blue denotes the specialist. Diagonal entries are in-distribution texts and off-diagonal entries are out-of-distribution texts. Texts are generated by ChatGPT.

Figure 17: Visualization of token-level routing results for `CoMiGS-1G1S` trained on AG News. Tokens are colored with the first expert choice at the 0th (first) layer. Orange denotes the generalist and blue denotes the specialist. Diagonal entries are in-distribution texts and off-diagonal entries are out-of-distribution texts. Texts are generated by ChatGPT.

Figure 18: Visualization of token-level routing results for `CoMiGS-1G1S` trained on AG News. Tokens are colored with the first expert choice at the 5th (middle) layer. Orange denotes the generalist and blue denotes the specialist. Diagonal entries are in-distribution texts and off-diagonal entries are out-of-distribution texts. Texts are generated by ChatGPT.

Figure 19: Visualization of token-level routing results for `CoMiGS-1G1S` trained on AG News. Tokens are colored with the first expert choice at the 11th (last) layer. Orange denotes the generalist and blue denotes the specialist. Diagonal entries are in-distribution texts and off-diagonal entries are out-of-distribution texts. Texts are generated by ChatGPT.

# F. Alternating Minimization Convergence

## F.1. Notation

Let us recall our notation from Sections 3.3 and 3.4. We have two differentiable functions $f_1(\Theta, \Phi) \equiv f_{\text{valid}}(\Theta, \Phi)$ and $f_2(\Theta, \Phi) \equiv f_{\text{train}}(\Theta, \Phi)$ that constitute our problem. For the sake of generality, let us assume that the target variables $\Theta$ and $\Phi$ belong to their corresponding feasible convex sets $Q$ and $\Omega$,

$$\Theta \in Q \subseteq \mathbb{R}^{|\Theta|}, \qquad \Phi \in \Omega \subseteq \mathbb{R}^{|\Phi|}.$$

Then, we consider the following *alternating minimization process*, starting from some initial $(\Theta_0, \Phi_0)$, for every $k \geq 0$:

$$\begin{aligned} \Phi_{k+1} &= \underset{\Phi \in \Omega}{\arg\min}\, f_1(\Theta_k, \Phi), \\ \Theta_{k+1} &= \underset{\Theta \in Q}{\arg\min}\, f_2(\Theta, \Phi_{k+1}). \end{aligned} \tag{12}$$

If $f_1 \equiv f_2$ that would be a standard alternation minimization as for minimizing one function $f_1$. However, in our setting $f_1$ and $f_2$ can be different.

For a fixed $\Theta$ and $\Phi$, let us denote the corresponding $\arg\min$ operators by

$$u_1(\Theta) := \underset{\Phi \in \Omega}{\arg\min}\, f_1(\Theta, \Phi)$$

and

$$u_2(\Phi) := \underset{\Theta \in Q}{\arg\min}\, f_2(\Theta, \Phi).$$

Using this notation, we can rewrite algorithm (12) as follows:

$$\Phi_{k+1} = u_1(\Theta_k), \qquad \Theta_{k+1} = u_2(\Phi_{k+1}), \qquad k \geq 0. \tag{13}$$

We further define the following operators:

$$\begin{aligned} T(\Theta) &:= u_2(u_1(\Theta)) \in Q, \qquad \Theta \in Q, \\ P(\Phi) &:= u_1(u_2(\Phi)) \in \Omega, \qquad \Phi \in \Omega. \end{aligned}$$

With this notation, we can rewrite the sequence $\{\Theta_k\}_{k \geq 0}$ simply as

$$\Theta_{k+1} = T(\Theta_k), \qquad k \geq 0. \tag{14}$$

We use the following **main assumption** on functions $f_1$ and $f_2$:

**Assumption 1.** *There exist $\Theta^\star \in Q$ and $\Phi^\star \in \Omega$ such that*

$$\Theta^\star = T(\Theta^\star) \quad and \quad \Phi^\star = P(\Phi^\star) \tag{15}$$

**Remark 1.** *Note that if $f_1 \equiv f_2 \equiv f$, condition (15) holds for the global minimizer of our function $(\Theta^\star, \Phi^\star) = \underset{\Theta \in Q, \Phi \in Q}{\arg\min}\, f(\Theta, \Phi)$.*

Clearly, this assumption should hold if functions $f_1$ and $f_2$ are *sufficiently close*: $f_1 \approx f_2$, or in case of overparametrized models. It remains an interesting open question: what are the general and joint conditions on $f_1$ and $f_2$ that imply (15).

## F.2. Contraction and Convergence

As we will see, it is natural to assume that operators $u_1$ and $u_2$ are *contractions*. We will provide a working example of our setting in the next section, where this condition will hold. We assume to have some norms fixed on $Q$ and $\Omega$, that are not necessarily Euclidean. For simplicity, and when it is clear from the context, we will use the same symbol $\|\cdot\|$ for both norms, even though they can be different for spaces of $\Theta$ and $\Phi$.

**Assumption 2.** *Let $u_1$ and $u_2$ be Lipschitz with some constants $\lambda_1, \lambda_2 > 0$:*

$$
\begin{aligned}
\|u_1(\boldsymbol{\Theta}) - u_1(\bar{\boldsymbol{\Theta}})\| &\leq \lambda_1 \|\boldsymbol{\Theta} - \bar{\boldsymbol{\Theta}}\|, &\forall \boldsymbol{\Theta}, \bar{\boldsymbol{\Theta}} \in Q, \\
\|u_2(\boldsymbol{\Phi}) - u_2(\bar{\boldsymbol{\Phi}})\| &\leq \lambda_2 \|\boldsymbol{\Phi} - \bar{\boldsymbol{\Phi}}\|, &\forall \boldsymbol{\Phi}, \bar{\boldsymbol{\Phi}} \in \Omega.
\end{aligned}
\tag{16}
$$

Under these assumptions we can show the convergence of the sequence $\{\boldsymbol{\Theta}_k\}_{k \geq 0}$ generated by (14). Indeed, for every $k \geq 0$, we have

$$
\|\boldsymbol{\Theta}_{k+1} - \boldsymbol{\Theta}^\star\| = \|T(\boldsymbol{\Theta}_k) - \boldsymbol{\Theta}^\star\| \overset{(15)}{=} \|T(\boldsymbol{\Theta}_k) - T(\boldsymbol{\Theta}^\star)\|
$$

$$
= \|u_2(u_1(\boldsymbol{\Theta}_k)) - u_2(u_1(\boldsymbol{\Theta}^\star))\| \overset{(6)}{\leq} \lambda_2 \|u_1(\boldsymbol{\Theta}_k) - u_1(\boldsymbol{\Theta}^\star)\| \overset{(6)}{\leq} \lambda_1 \lambda_2 \|\boldsymbol{\Theta}_k - \boldsymbol{\Theta}^\star\|,
$$

and we see that $\boldsymbol{\Theta}_k \to \boldsymbol{\Theta}^\star$ with the linear rate. The same reasoning can be applied to the sequence $\{\boldsymbol{\Phi}_k\}_{k \geq 1}$. Thus, we have established the following general convergence result.

**Theorem F.1** (Theorem 3.1). *Let Assumptions 1, 2 hold and $\lambda_1 \cdot \lambda_2 < 1$. Then, the sequence $(\boldsymbol{\Theta}_k, \boldsymbol{\Phi}_k)_{k \geq 0}$ generated by alternating process (12) converges to $(\boldsymbol{\Theta}^\star, \boldsymbol{\Phi}^\star)$ linearly, for every $k \geq 0$:*

$$
\begin{aligned}
\|\boldsymbol{\Theta}_k - \boldsymbol{\Theta}^\star\| &\leq (\lambda_1 \lambda_2)^k \|\boldsymbol{\Theta}_0 - \boldsymbol{\Theta}^\star\|, \\
\|\boldsymbol{\Phi}_k - \boldsymbol{\Phi}^\star\| &\leq (\lambda_1 \lambda_2)^k \|\boldsymbol{\Phi}_0 - \boldsymbol{\Phi}^\star\|.
\end{aligned}
\tag{17}
$$

**Example 1.** *Consider the following quadratic objective*

$$
f(\boldsymbol{\Theta}, \boldsymbol{\Phi}) = \tfrac{1}{2}\langle \boldsymbol{A}\boldsymbol{\Theta}, \boldsymbol{\Theta}\rangle + \tfrac{1}{2}\langle \boldsymbol{B}\boldsymbol{\Phi}, \boldsymbol{\Phi}\rangle + \langle \boldsymbol{C}\boldsymbol{\Theta}, \boldsymbol{\Phi}\rangle,
$$

*where $\boldsymbol{A} = \boldsymbol{A}^\top \in \mathbb{R}^{|\boldsymbol{\Theta}| \times |\boldsymbol{\Theta}|}$ and $\boldsymbol{B} = \boldsymbol{B}^\top \in \mathbb{R}^{|\boldsymbol{\Phi}| \times |\boldsymbol{\Phi}|}$ are symmetric matrices, and $\boldsymbol{C} \in \mathbb{R}^{|\boldsymbol{\Phi}| \times |\boldsymbol{\Theta}|}$. We assume that $f$ is strictly convex, which means*

$$
\boldsymbol{H} = \begin{bmatrix} \boldsymbol{A} & \boldsymbol{C}^\top \\ \boldsymbol{C} & \boldsymbol{B} \end{bmatrix} \succ \boldsymbol{0}.
$$

*Clearly, for this objective, we have $\boldsymbol{\Theta}^\star = \boldsymbol{0}$ and $\boldsymbol{\Phi}^\star = \boldsymbol{0}$. Then*

$$
u_1(\boldsymbol{\Theta}) := \underset{\boldsymbol{\Phi}}{\arg\min} \, f(\boldsymbol{\Theta}, \boldsymbol{\Phi}) = -\boldsymbol{B}^{-1}\boldsymbol{C}\boldsymbol{\Theta} \quad \text{and}
$$

$$
u_2(\boldsymbol{\Phi}) := \underset{\boldsymbol{\Theta}}{\arg\min} \, f(\boldsymbol{\Theta}, \boldsymbol{\Phi}) = -\boldsymbol{A}^{-1}\boldsymbol{C}^\top\boldsymbol{\Phi}.
$$

*Hence, the composition operator $T := u_2 \circ u_1$ is linear:*

$$
T(\boldsymbol{\Theta}) = \boldsymbol{A}^{-1}\boldsymbol{C}^\top\boldsymbol{B}^{-1}\boldsymbol{C}\boldsymbol{\Theta},
\tag{18}
$$

*and it holds*

$$
\|T(\boldsymbol{\Theta}) - \boldsymbol{\Theta}^\star\| \leq \|\boldsymbol{A}^{-1}\boldsymbol{C}^\top\boldsymbol{B}^{-1}\boldsymbol{C}\| \cdot \|\boldsymbol{\Theta} - \boldsymbol{\Theta}^\star\|.
$$

*Now, denoting by $\mu > 0$ and $L \geq \mu$ the smallest and the largest eigenvalues of matrix $\boldsymbol{H}$ correspondingly, and using the Schur complement, we conclude that*

$$
\mu\boldsymbol{I} \preceq \boldsymbol{A} \preceq L\boldsymbol{I}, \quad \text{and} \quad \mu\boldsymbol{I} \preceq \boldsymbol{A} - \boldsymbol{C}^\top\boldsymbol{B}^{-1}\boldsymbol{C} \preceq L\boldsymbol{I},
\tag{19}
$$

*from which we are able to bound the norm of our matrix as follows:*

$$
\|\boldsymbol{A}^{-1}\boldsymbol{C}^\top\boldsymbol{B}^{-1}\boldsymbol{C}\| = \|\boldsymbol{A}^{-1/2}(\boldsymbol{C}^\top\boldsymbol{B}^{-1}\boldsymbol{C})\boldsymbol{A}^{-1/2}\| \overset{(19)}{\leq} \frac{L-\mu}{L} < 1,
$$

*which proves the contraction property.* $\square$

**Example 2.** *Note that for a general differentiable function $f$, using the Taylor expansion, the operator $T = u_2 \circ u_1$, where $u_1(\mathbf{\Theta}) := \arg\min_{\mathbf{\Phi}} f(\mathbf{\Theta}, \mathbf{\Phi})$ and $u_2(\mathbf{\Phi}) := \arg\min_{\mathbf{\Theta}} f(\mathbf{\Theta}, \mathbf{\Phi})$, can be expressed as follows (compare with (18)):*

$$T(\mathbf{\Theta}) - \mathbf{\Theta}^\star \;\; = \;\; \boldsymbol{H}_{11}^{-1}\boldsymbol{H}_{12}\boldsymbol{H}_{22}^{-1}\boldsymbol{H}_{21}(\mathbf{\Theta} - \mathbf{\Theta}^\star),$$

*where*

$$\boldsymbol{H}_{11} \;\; = \;\; \int_0^1 \frac{\partial^2 f}{\partial \mathbf{\Theta}^2}(\mathbf{\Theta}^\star + \tau(T(\mathbf{\Theta}) - \mathbf{\Theta}^\star), \mathbf{\Phi}^\star + \tau(u_1(\mathbf{\Theta}) - \mathbf{\Phi}^\star))d\tau,$$

$$\boldsymbol{H}_{12} \;\; = \;\; \int_0^1 \frac{\partial^2 f}{\partial \mathbf{\Theta}\partial \mathbf{\Phi}}(\mathbf{\Theta}^\star + \tau(T(\mathbf{\Theta}) - \mathbf{\Theta}^\star), \mathbf{\Phi}^\star + \tau(u_1(\mathbf{\Theta}) - \mathbf{\Phi}^\star))d\tau,$$

$$\boldsymbol{H}_{22} \;\; = \;\; \int_0^1 \frac{\partial^2 f}{\partial \mathbf{\Phi}^2}(\mathbf{\Theta}^\star + \tau(\mathbf{\Theta} - \mathbf{\Theta}^\star), \mathbf{\Phi}^\star + \tau(u_1(\mathbf{\Theta}) - \mathbf{\Phi}^\star))d\tau,$$

$$\boldsymbol{H}_{21} \;\; = \;\; \int_0^1 \frac{\partial^2 f}{\partial \mathbf{\Phi}\partial \mathbf{\Theta}}(\mathbf{\Theta}^\star + \tau(\mathbf{\Theta} - \mathbf{\Theta}^\star), \mathbf{\Phi}^\star + \tau(u_1(\mathbf{\Theta}) - \mathbf{\Phi}^\star))d\tau.$$

*Therefore, assuming that the Hessian is strictly positive definite and Lipschitz continuous in a neighborhood of the solution, localizing the current point to the neighborhood, $\mathbf{\Theta} \approx \mathbf{\Theta}^\star$ and $\mathbf{\Phi} \approx \mathbf{\Phi}^\star$, we can obtain the contraction property, as in the previous example (see, e.g., Theorem 1.2.5 in (Nesterov, 2018) for the local analysis of Newton's method).*

### F.3. Linear Modeling and Decoupling

In this section, let us study an important example of *linear models*, applicable to both experts and the router. As we will show, in this case and under very mild assumptions we can justify all conditions from the previous section and therefore obtain the global linear convergence for our alternating process.

**Problem Formulation**   For simplicity, we consider the case of one client and assume that training and validation datasets are the same, $\boldsymbol{X}^{\text{train}} = \boldsymbol{X}^{\text{valid}}$. However, our observations can be generalized to a more general case of several clients, and different but statistically similar datasets $\boldsymbol{X}^{\text{train}} \sim \boldsymbol{X}^{\text{valid}}$. Hence, we have, $f_1 \equiv f_2 \equiv f$. Note that in this case, our bi-level formulation is also equivalent to joint minimization of $f$ w.r.t. all variables.

We assume that our client has one generalist expert model, that we denote by $\boldsymbol{\theta}^0 \in \mathbb{R}^d$, and $N \geq 0$ specialist experts, that we denote by $\boldsymbol{\theta}^1, \ldots \boldsymbol{\theta}^N \in \mathbb{R}^d$. We compose these models together as matrix $\mathbf{\Theta} = (\boldsymbol{\theta}^0, \ldots, \boldsymbol{\theta}^N)$. In principle, different models can have different expressivity, which we take into account in our modeling by a convex set of constraints: $\mathbf{\Theta} \in Q \subseteq \mathbb{R}^{d \times (N+1)}$.

We denote by $\boldsymbol{\phi}^0, \ldots, \boldsymbol{\phi}^N \in \mathbb{R}^d$ the parameters of our Router, composed together as matrix $\mathbf{\Phi} = (\boldsymbol{\phi}^0, \ldots, \boldsymbol{\phi}^N)$, which can also be constrained by a convex set: $\mathbf{\Phi} \in \Omega \subseteq \mathbb{R}^{d \times (N+1)}$. For a given data input $\boldsymbol{x} \in \mathbb{R}^d$, the Router decides which experts to use with the SoftMax operation $\boldsymbol{x} \mapsto \boldsymbol{\pi}_{\mathbf{\Phi}}(\boldsymbol{x}) \in \Delta_N$, where

$$\Delta_N \;\; := \;\; \left\{ \boldsymbol{y} \in \mathbb{R}_+^{N+1} \; : \; \sum_{j=0}^N y^{(j)} = 1 \right\}$$

is the standard Simplex, and

$$\pi_{\mathbf{\Phi}}^{(j)}(\boldsymbol{x}) \;\; := \;\; \frac{\exp(\langle \boldsymbol{\phi}^j, \boldsymbol{x}\rangle)}{\sum_{k=0}^N \exp(\langle \phi^k, \boldsymbol{x}\rangle)}. \tag{20}$$

Under these assumptions, we set the following structure of our optimization objective,

$$f(\mathbf{\Theta}, \mathbf{\Phi}) \;\; = \;\; \frac{1}{n} \sum_{i=1}^n \ell_i\left( \sum_{j=0}^N \pi_{\mathbf{\Phi}}^{(j)}(\boldsymbol{x}_i) \cdot \langle \boldsymbol{\theta}^j, \boldsymbol{x}_i \rangle \right) + \frac{\alpha}{2}\left( \|\mathbf{\Theta}\|_F^2 + \|\mathbf{\Phi}\|_F^2 \right), \tag{21}$$

where $\boldsymbol{x}_1, \ldots, \boldsymbol{x}_n$ are given data vectors, and $\ell_i(\cdot), 1 \leq i \leq n$ are the corresponding convex losses (e.g. the logistic loss for binary classification, or the quadratic loss for regression problem). We use $\alpha \geq 0$ as a regularization parameter, which can also be seen as the *weight decay*, and $\|\cdot\|_F$ is the Frobenius norm of a matrix.

**Decoupling**   Let us introduce *the auxiliary variables*, $\boldsymbol{\lambda}^i \in \Delta_N$, $1 \le i \le n$, and $\boldsymbol{\Lambda} = (\boldsymbol{\lambda}^1, \ldots, \boldsymbol{\lambda}^n) \in \Delta_N^n \subseteq \mathbb{R}^{(N+1)\times n}$, which is a column-stochastic matrix. Employing the matrix notation, we can rewrite our problem in the following form:

$$\min_{\substack{\boldsymbol{\Theta}\in Q, \boldsymbol{\Phi}\in\Omega \\ \boldsymbol{\Lambda}\in\Delta_N^n}} \left\{ \frac{1}{n}\sum_{i=1}^n \ell_i\Big(\langle \boldsymbol{\lambda}^i, \boldsymbol{\Theta}^\top \boldsymbol{x}_i \rangle\Big) + \frac{\alpha}{2}\Big(\|\boldsymbol{\Theta}\|_F^2 + \|\boldsymbol{\Phi}\|_F^2\Big) \ : \ \boldsymbol{\lambda}^i = \pi_{\boldsymbol{\Phi}}(\boldsymbol{x}_i), \ 1\le i\le n \right\}. \tag{22}$$

Now, we apply the relaxation of constrained problem (22) by the following *decouple* of $\lambda^i$ from $\pi_{\boldsymbol{\Phi}}(\boldsymbol{x}_i)$, with some parameter $\mu \ge 0$ and a *distance function* $V : \Delta_N \times \Delta_N \to \mathbb{R}_+$ between distributions:

$$\min_{\substack{\boldsymbol{\Theta}\in Q, \boldsymbol{\Phi}\in\Omega \\ \boldsymbol{\Lambda}\in\Delta_N^n}} \left\{ F_\mu(\Theta, \boldsymbol{\Phi}, \boldsymbol{\Lambda}) \ := \ \frac{1}{n}\sum_{i=1}^n \ell_i\Big(\langle \boldsymbol{\lambda}^i, \boldsymbol{\Theta}^\top \boldsymbol{x}_i \rangle\Big) + \frac{\alpha}{2}\Big(\|\boldsymbol{\Theta}\|_F^2 + \|\boldsymbol{\Phi}\|_F^2\Big) + \frac{\mu}{2n}\sum_{i=1}^n V(\boldsymbol{\lambda}^i; \boldsymbol{\pi}_{\boldsymbol{\Phi}}(\boldsymbol{x}_i)) \right\}. \tag{23}$$

A natural choice for $V$ is the *Kullback–Leibler divergence*, which gives, for every $1 \le i \le n$:

$$V(\boldsymbol{\lambda}^i; \boldsymbol{\pi}_{\boldsymbol{\Phi}}(\boldsymbol{x}_i)) \ := \ \sum_{j=0}^N \big[\boldsymbol{\lambda}^i\big]^{(j)} \ln\big[\boldsymbol{\lambda}^i\big]^{(j)} - \sum_{j=0}^N \big[\boldsymbol{\lambda}^i\big]^{(j)} \ln\big[\boldsymbol{\pi}_{\boldsymbol{\Phi}}(\boldsymbol{x}_i)\big]^{(j)}$$

$$\overset{(20)}{=} \ \sum_{j=0}^N \big[\boldsymbol{\lambda}^i\big]^{(j)} \Big(\ln\big[\boldsymbol{\lambda}^i\big]^{(j)} - \langle \boldsymbol{\phi}^j, \boldsymbol{x}_i \rangle\Big) + \ln\Big(\sum_{j=0}^N \exp\big(\langle \boldsymbol{\phi}^j, \boldsymbol{x}_i \rangle\big)\Big)$$

$$= \ d(\boldsymbol{\lambda}^i) - \langle \boldsymbol{\lambda}^i, \boldsymbol{\Phi}^\top \boldsymbol{x}_i \rangle + s(\boldsymbol{\Phi}^\top \boldsymbol{x}_i),$$

where

$$d(\boldsymbol{\lambda}) \ := \ \sum_{j=0}^N \lambda^{(j)} \ln \lambda^{(j)}, \qquad \boldsymbol{\lambda} \in \Delta_N,$$

is the negative entropy, and

$$s(\boldsymbol{y}) \ := \ \ln\Big(\sum_{j=0}^N \exp y^{(j)}\Big), \qquad \boldsymbol{y} \in \mathbb{R}^{N+1}$$

is the log-sum-exp function. Note that both $d(\cdot)$ and $s(\cdot)$ are convex functions on their domains. Moreover, it is well known that $d(\cdot)$ is *strongly convex* w.r.t. $\ell_1$-norm (see, e.g., Example 2.1.2 in (Nesterov, 2018)):

$$\langle \nabla^2 d(\boldsymbol{\lambda})\boldsymbol{h}, \boldsymbol{h} \rangle \ \ge \ \|\boldsymbol{h}\|_1^2, \qquad \boldsymbol{\lambda} \in \Delta_N, \boldsymbol{h} \in \mathbb{R}^{N+1}. \tag{24}$$

Therefore, we obtain the following **decoupled optimization formulation**:

$$\boxed{\min_{\substack{\boldsymbol{\Theta}\in Q, \boldsymbol{\Phi}\in\Omega \\ \boldsymbol{\Lambda}\in\Delta_N^n}} \left\{ \begin{aligned} & F_\mu(\boldsymbol{\Theta}, \boldsymbol{\Phi}, \boldsymbol{\Lambda}) \\ & = \frac{1}{n}\sum_{i=1}^n \Big[\ell_i\Big(\langle \boldsymbol{\lambda}^i, \boldsymbol{\Theta}^\top \boldsymbol{x}_i \rangle\Big) + \mu\Big(d(\boldsymbol{\lambda}^i) + s(\boldsymbol{\Phi}^\top \boldsymbol{x}_i) - \langle \boldsymbol{\lambda}^i, \boldsymbol{\Phi}^\top \boldsymbol{x}_i \rangle\Big)\Big] + \frac{\alpha}{2}\Big(\|\boldsymbol{\Theta}\|_F^2 + \|\boldsymbol{\Phi}\|_F^2\Big) \end{aligned} \right\}.} \tag{25}$$

It is clear that setting parameter $\mu := +\infty$, we obtain that (25) is equivalent to our original problem (22). However, for $\mu < +\infty$ we obtain more flexible formulation with auxiliary distributions $\boldsymbol{\lambda}^i \in \Delta_N$, each for every data sample $1 \le i \le n$, that makes it easier to treat the problem. Parameters $(\boldsymbol{\lambda}^i)$ has an interpretation of *latent variables*, which makes our approach similar to the classical EM-algorithm (Jordan & Jacobs, 1994). We note a similar work from Almansoori et al. (2024), which proposes to train a mixture of generalists on local routers, which resembles a simplified version of our method.

It is clear that function $F_\mu(\boldsymbol{\Theta}, \boldsymbol{\Phi}, \boldsymbol{\Lambda})$ is *partially convex*: it is convex w.r.t $(\boldsymbol{\Theta}, \boldsymbol{\Phi})$ when $\boldsymbol{\Lambda}$ is fixed, and it is also convex w.r.t. $\boldsymbol{\Lambda}$ when $(\boldsymbol{\Theta}, \boldsymbol{\Phi})$ is fixed.

In what follows, we show that under very mild conditions and choosing regularization parameter $\alpha, \mu \ge 0$ sufficiently large, we can ensure that $F_\mu(\cdot)$ is *jointly strongly convex*, regardless of non-convex cross terms: $\ell_i\big(\langle \boldsymbol{\lambda}^i, \boldsymbol{\Theta}^\top \boldsymbol{x}_i \rangle\big)$ and $\langle \boldsymbol{\lambda}^i, \boldsymbol{\Phi}^\top \boldsymbol{x}_i \rangle$. Our theory generalizes a recent approach to soft clustering (Nesterov, 2020). With this technique, we will be able to show the global linear convergence rate for the alternating minimization approach that we discussed in the previous sections.

**Joint Strong Convexity** Let us consider the $i$-th term of our objective (25) that correspond to the data sample with index $1 \leq i \leq n$. Omitting extra indices, we obtain the following function,

$$F(\boldsymbol{\Theta}, \boldsymbol{\Phi}, \boldsymbol{\lambda}) \quad = \quad \ell\Big(\langle\boldsymbol{\lambda}, \boldsymbol{\Theta}^\top \boldsymbol{x}\rangle\Big) - \mu\langle\boldsymbol{\lambda}, \boldsymbol{\Phi}^\top \boldsymbol{x}\rangle + \tfrac{\alpha}{2}\Big(\|\boldsymbol{\Theta}\|_F^2 + \|\boldsymbol{\Phi}\|_F^2\Big) + \mu d(\boldsymbol{\lambda}) + \mu s(\boldsymbol{\Phi}^\top \boldsymbol{x}), \tag{26}$$

where $\boldsymbol{\Theta} \in Q$, $\boldsymbol{\Phi} \in \Omega$, $\boldsymbol{\lambda} \in \Delta_N$. Our goal is to ensure that (26) is strongly convex w.r.t to the standard Euclidean norm of the joint variable. Namely, we establish the following result.

**Proposition 1.** *Let the loss function $\ell(\cdot)$ be convex and assume that its first derivative is bounded: $\rho \geq \max_t \ell'(t)$. Assume that the regularization coefficient is sufficiently large:*

$$\alpha \quad \geq \quad 2\|\boldsymbol{x}\|^2 \max\big\{\mu, \tfrac{\rho^2}{\mu}\big\}. \tag{27}$$

*Then the objective in (26) is strongly convex.*

*Proof.* Note that our objective (26) can be separated in $\boldsymbol{\Phi}$ and $\boldsymbol{\Theta}$. We construct the following functions $g_1$ and $g_2$.

$$g_1(\boldsymbol{\Theta}, \boldsymbol{\lambda}) \quad := \quad \ell\Big(\langle\boldsymbol{\lambda}, \boldsymbol{\Theta}^\top \boldsymbol{x}\rangle\Big) + \tfrac{\alpha}{4}\|\boldsymbol{\Theta}\|_F^2 + \tfrac{\mu}{4}d(\boldsymbol{\lambda}) \quad \text{and} \quad g_2(\boldsymbol{\Phi}, \boldsymbol{\lambda}) \quad := \quad -\mu\langle\boldsymbol{\lambda}, \boldsymbol{\Phi}^\top \boldsymbol{x}\rangle + \tfrac{\alpha}{4}\|\boldsymbol{\Phi}\|_F^2 + \tfrac{\mu}{4}d(\boldsymbol{\lambda}) \ ,$$

and $F(\boldsymbol{\Theta}, \boldsymbol{\Phi}, \boldsymbol{\lambda}) \equiv g_1(\boldsymbol{\Theta}, \boldsymbol{\lambda}) + g_2(\boldsymbol{\Phi}, \boldsymbol{\lambda}) + \tfrac{\alpha}{4}\Big(\|\boldsymbol{\Theta}\|_F^2 + \|\boldsymbol{\Phi}\|_F^2\Big) + \tfrac{\mu}{2}d(\boldsymbol{\lambda}) + \mu s(\boldsymbol{\Phi}^\top \boldsymbol{x})$. Since the log-sum-exp function $s(\cdot)$ is strictly convex, and the negative entropy $d(\cdot)$, as well as the Frobenius norm, are strongly convex, it suffices to prove convexity for $g_1$ and $g_2$. Computing the second derivative of $g_1$ and applying it to an arbitrary direction $\boldsymbol{z} = [\boldsymbol{H}; \boldsymbol{h}]$ of corresponding shapes, we get

$$\begin{aligned}
\langle\nabla^2 g_1(\boldsymbol{\Theta}, \boldsymbol{\lambda})\boldsymbol{z}, \boldsymbol{z}\rangle \quad &= \quad \tfrac{\alpha}{2}\|\boldsymbol{H}\|_F^2 + \tfrac{\mu}{4}\langle\nabla^2 d(\boldsymbol{\lambda})\boldsymbol{h}, \boldsymbol{h}\rangle \\[4pt]
&\quad + \ell''(\langle\boldsymbol{\lambda}, \boldsymbol{\Theta}^\top \boldsymbol{x}\rangle) \cdot \underbrace{\Big[\langle\boldsymbol{h}, \boldsymbol{\Theta}^\top \boldsymbol{x}\rangle^2 + \langle\boldsymbol{\lambda}, \boldsymbol{H}^\top \boldsymbol{x}\rangle^2 + \langle\boldsymbol{h}, \boldsymbol{\Theta}^\top \boldsymbol{x}\rangle \cdot \langle\boldsymbol{\lambda}, \boldsymbol{H}^\top \boldsymbol{x}\rangle\Big]}_{\geq 0} \\[4pt]
&\quad + \ell'(\langle\boldsymbol{\lambda}, \boldsymbol{\Theta}^\top \boldsymbol{x}\rangle) \cdot \langle\boldsymbol{h}, \boldsymbol{H}^\top \boldsymbol{x}\rangle \\[6pt]
&\geq \quad \tfrac{\alpha}{2}\|\boldsymbol{H}\|_F^2 + \tfrac{\mu}{4}\|\boldsymbol{h}\|_1^2 - \rho\|\boldsymbol{h}\|_1 \cdot \|\boldsymbol{H}\|_F \cdot \|\boldsymbol{x}\| \\[6pt]
&\overset{(*)}{\geq} \quad \|\boldsymbol{h}\|_1 \cdot \|\boldsymbol{H}\|_F \cdot \Big(\sqrt{\tfrac{\alpha\mu}{2}} - \rho\|\boldsymbol{x}\|\Big) \overset{(27)}{\geq} \quad 0,
\end{aligned}$$

where we used Young's inequality in $(*)$. The bound for $g_2$ follows by the same reasoning, substituting $\ell(t) := \mu t$ and therefore setting $\rho := \mu$. $\qquad\square$

For the decoupled optimization formulation (25) it is natural to organize iterations in the following sequential order, starting from an arbitrary $\boldsymbol{\Theta}_0 \in Q$ and $\boldsymbol{\Phi}_0 \in \Omega$, for some $\mu > 0$:

$$\boxed{\begin{aligned}
\boldsymbol{\Lambda}_{k+1} \quad &= \quad \underset{\boldsymbol{\Lambda} \in \Delta_N^n}{\arg\min}\, F_\mu(\boldsymbol{\Theta}_k, \boldsymbol{\Phi}_k, \boldsymbol{\Lambda}), \\
\boldsymbol{\Phi}_{k+1} \quad &= \quad \underset{\boldsymbol{\Phi} \in \Omega}{\arg\min}\, F_\mu(\boldsymbol{\Theta}_k, \boldsymbol{\Phi}, \boldsymbol{\Lambda}_{k+1}), \\
\boldsymbol{\Theta}_{k+1} \quad &= \quad \underset{\boldsymbol{\Theta} \in Q}{\arg\min}\, F_\mu(\boldsymbol{\Theta}, \boldsymbol{\Phi}_{k+1}, \boldsymbol{\Lambda}_{k+1}).
\end{aligned}} \tag{28}$$

Note that each minimization subproblem in (28) is convex and can be implemented very efficiently by means of linear algebra and convex optimization. At the same time, due to decoupling of variables and strong convexity we are able to ensure the global convergence of this process to the solution of (25).

### F.4. Convergence for Functional Residual

Note that in our decoupled optimization formulation (25), variables $\boldsymbol{\Theta}$ and $\boldsymbol{\Phi}$ are independent of each other, when $\boldsymbol{\Lambda}$ is fixed. Therefore, the second and third step in iteration process (28) can be done independently.

For the sake of notation, let us denote $\boldsymbol{X} \equiv \boldsymbol{\Lambda}$, concatenated variable $\boldsymbol{Y} \equiv (\boldsymbol{\Theta}, \boldsymbol{\Phi})$, and the objective in new variables as $f(\boldsymbol{X}, \boldsymbol{Y}) \equiv F_\mu(\boldsymbol{\Theta}, \boldsymbol{\Phi}, \boldsymbol{\Lambda})$. By our previous analysis, we can assume that $f$ is strongly convex. We denote by $\mu$ the parameter of strong convexity and by $L$ the constant of Lipschitz continuity of the gradient of $f$. Its global minimum is denoted by $(\boldsymbol{X}^\star, \boldsymbol{Y}^\star)$, and correspondingly $f^\star := f(\boldsymbol{X}^\star, \boldsymbol{Y}^\star)$.

Then, iteration process (28) can be rewritten simply as the following alternating iterations, for $k \geq 0$:

$$\boldsymbol{Y}_{k+1} = \arg\min_{\boldsymbol{Y} \in \mathcal{Y}} f(\boldsymbol{X}_k, \boldsymbol{Y}),$$

$$\boldsymbol{X}_{k+1} = \arg\min_{\boldsymbol{X} \in \mathcal{X}} f(\boldsymbol{X}, \boldsymbol{Y}_{k+1}),$$

where $\mathcal{X}$ and $\mathcal{Y}$ are the corresponding convex domains ($\mathcal{X} \equiv \Delta_N^n$ and $\mathcal{Y} \equiv \Omega \times Q$).

Then, the stationary condition for $\boldsymbol{Y}_{k+1}$ (see, e.g., Theorem 3.1.23 in (Nesterov, 2018)) gives

$$\langle \tfrac{\partial f}{\partial \boldsymbol{Y}}(\boldsymbol{X}_k, \boldsymbol{Y}_{k+1}), \boldsymbol{Y} - \boldsymbol{Y}_{k+1} \rangle \geq 0, \qquad \forall \boldsymbol{Y} \in \mathcal{Y}. \tag{29}$$

Choosing

$$\gamma := \tfrac{\mu}{L} \leq 1, \tag{30}$$

we obtain

$$\gamma f(\boldsymbol{X}^\star, \boldsymbol{Y}^\star) + (1 - \gamma) f(\boldsymbol{X}_k, \boldsymbol{Y}_{k+1})$$

$$\overset{(*)}{\geq} \gamma \Big[ f(\boldsymbol{X}_k, \boldsymbol{Y}_{k+1}) + \langle \tfrac{\partial f}{\partial \boldsymbol{X}}(\boldsymbol{X}_k, \boldsymbol{Y}_{k+1}), \boldsymbol{X}^\star - \boldsymbol{X}_k \rangle + \langle \tfrac{\partial f}{\partial \boldsymbol{Y}}(\boldsymbol{X}_k, \boldsymbol{Y}_{k+1}), \boldsymbol{Y}^\star - \boldsymbol{Y}_{k+1} \rangle + \tfrac{\mu}{2}\|\boldsymbol{X}^\star - \boldsymbol{X}_k\|^2 \Big]$$

$$+ (1 - \gamma) f(\boldsymbol{X}_k, \boldsymbol{Y}_{k+1})$$

$$\overset{(29),(30)}{\geq} f(\boldsymbol{X}_k, \boldsymbol{Y}_{k+1}) + \langle \tfrac{\partial f}{\partial \boldsymbol{X}}(\boldsymbol{X}_k, \boldsymbol{Y}_{k+1}), \gamma(\boldsymbol{X}^\star - \boldsymbol{X}_k) \rangle + \tfrac{L}{2}\|\gamma(\boldsymbol{X}^\star - \boldsymbol{X}_k)\|^2$$

$$\geq \min_{\boldsymbol{X} \in \mathcal{X}} \Big\{ f(\boldsymbol{X}_k, \boldsymbol{Y}_{k+1}) + \langle \tfrac{\partial f}{\partial \boldsymbol{X}}(\boldsymbol{X}_k, \boldsymbol{Y}_{k+1}), \boldsymbol{X} - \boldsymbol{X}_k \rangle + \tfrac{L}{2}\|\boldsymbol{X} - \boldsymbol{X}_k\|^2 \Big\}$$

$$\overset{(**)}{\geq} \min_{\boldsymbol{X} \in \mathcal{X}} \{ f(\boldsymbol{X}, \boldsymbol{Y}_{k+1}) \} = f(\boldsymbol{X}_{k+1}, \boldsymbol{Y}_{k+1}),$$

where in $(*)$ we used strong convexity, and in $(**)$ we used the Lipschitz continuity of the gradient. Thus, we get the following inequality:

$$f(\boldsymbol{X}_{k+1}, \boldsymbol{Y}_{k+1}) - f^\star \leq (1 - \gamma)\Big( f(\boldsymbol{X}_k, \boldsymbol{Y}_{k+1}) - f^\star \Big),$$

and using the same reasoning for $\boldsymbol{Y}_k \mapsto \boldsymbol{Y}_{k+1}$ update, we obtain

$$f(\boldsymbol{X}_{k+1}, \boldsymbol{Y}_{k+1}) - f^\star \leq (1 - \gamma)^2 \Big( f(\boldsymbol{X}_k, \boldsymbol{Y}_k) - f^\star \Big),$$

which is the global linear rate. Thus, we have established formally the following convergence result.

**Theorem F.2.** *Let $f$ be strongly convex with constant $\mu > 0$, and let its gradient be Lipschitz continuous with constant $L > 0$. Then, for $k \geq 0$ iteration of the alternating minimization process, we have*

$$f(\boldsymbol{X}_k, \boldsymbol{Y}_k) - f^\star \leq \Big(1 - \tfrac{\mu}{L}\Big)^{2k} \Big( f(\boldsymbol{X}_0, \boldsymbol{Y}_0) - f^\star \Big).$$

Note that this result is directly applicable for our linear models from previous sections, $f(\boldsymbol{X}, \boldsymbol{Y}) \equiv F_\mu(\boldsymbol{\Theta}, \boldsymbol{\Phi}, \boldsymbol{\Lambda})$, as we show that objective (25) is jointly strongly convex, when the regularization parameter is sufficiently large.

