# OpenReview forum: "On-Device Collaborative Language Modeling via a Mixture of Generalists and Specialists"
_ICML.cc/2025/Conference — ICML 2025 poster_

### Official Review · Reviewer_TG8c · 2025-03-04

**Overall Recommendation:** 4

**Summary:**

The paper introduces a method for training language models collaboratively between multiple devices/clients and personalizing them to their on-device data at the same time. This is done by introducing a mixture of generalist experts and specialist experts, where the generalists are trained collaboratively (e.g., through federated averaging) and the specialists are trained locally (without aggregation). The routers are trained locally as well. Thus, only the generalist experts are aggregated and learned collaboratively across clients, which reduces communication overhead. The problem is formulated as a bi-level optimization problem with a training algorithm that optimizes the experts on a training dataset and then optimizes the routers on a validation dataset, done alternatively. This method addresses both data heterogeneity and computational resources heterogeneity, which is purportedly the first to accomplish that , and it also separates model heterogeneity from data quantity as shown in the extensive experiments section. The authors also provide convergence analysis under some standard assumptions.

## update after rebuttal

The authors have addressed most of my concerns and I'm satisfied with the rebuttal. I will keep my original positive rating.

**Claims And Evidence:**

The authors claim that the bi-level formulation of the MoE learning objective and the alternating minimization algorithm are new. The formulation itself might be, but an alternating minimization algorithm of the router and the experts is quite general, so it is difficult to claim that it is "new", especially since it resembles expectation maximization (which the authors themselves mention). For example, [this work](https://arxiv.org/abs/2410.03497) trains mixture of "generalists" on local (input-independent) routers, which resembles a simplified version of the proposed method.

The theoretical analysis is interesting and shows linear convergence. The assumptions are claimed to be "suitable". I agree that the claims are standard. However, I would argue that the existence of a minimizer might not be necessarily be a suitable assumption in language modeling. In general, the loss doesn't go to 0 on these problems.

The claim regarding not overfitting even with many experts and little data might only be applicable because this is a fine-tuning problem, but it is still an interesting result.

The other claims are true and are well corroborated in the experiments, though I'm not sure whether this work is, indeed, the first that addresses both data heterogeneity and computational resources heterogeneity.

**Essential References Not Discussed:**

I do not know of any essential references related to this work that were not discussed.

**Experimental Designs Or Analyses:**

The experimental design does capture interesting practical scenarios. For example, the out-of-distribution experiments uses a validation and test set that is from a different distribution from the training dataset, so that assumption 1 is violated. However, the method still performs favorably, which shows its robustness. Other experiments also provide ample evidence for the claims in the contributions list. The authors run experiments on various datasets and compare to many recent baselines. The analysis is also valid for the considered setup.

**Methods And Evaluation Criteria:**

The proposed method is very sound and the authors provide theoretical and experimental evidence that it works.
The  authors provide extensive experimentation and ablation studies to evaluate the method, all of which show positive evidence that the proposed method shows good performance, robustness, and efficiency with respect to the baselines.

**Other Comments Or Suggestions:**

In line 1435, the authors mention that the goal is to show that (26) is strongly convex, but the next line closes the section by saying that direct computation of the Hessian shows the joint strong convexity. Perhaps the author can write the details to make it clearer to the reader how it holds for sufficiently large $\alpha$ and $\mu$.

Minor comment: the simplex defined in line 1361 should actually be $\Delta^N$ because there are $N$ degrees of freedom/dimensions for $N+1$ values (the last value is completely determined by the rest).

Typos: There is a little 'z' hanging on the right side of Table 2. Also, in line 1406, there is a typo: "srongly".

**Other Strengths And Weaknesses:**

This work provides a practical and theoretically well-grounded method for an important application: collaborative training of personalized language models. The solution is sound and the experiments show good performance.

A great strength of this paper is that they share well-written code for reproducing the experimental results.

One weakness might be that this method mainly works for fine-tuning. It would be interesting to see whether the same benefits still hold for full training. Another could be that this procedure is only applicable to mixture of experts that routes tokens, i.e., transformer-based models and tokenized data.

**Questions For Authors:**

When you have many generalists, and one device can only use a few, how do you choose the generalists which will be assigned to those low-resource devices?

**Relation To Broader Scientific Literature:**

The contributions of this paper are important to the literature. They tackle multiple issues of interest. For example, the data heterogeneity and systems heterogeneity in federated learning. the proposed method is also relevant to practical applications of LLMs in the real world.

**Theoretical Claims:**

I checked the correctness of the proofs in the Appendix. The analysis is strightforward and clear. I did not find any significant errors.

---

> ### Author Rebuttal · Authors · 2025-04-01
>
> Thank you very much for your time and expertise in reviewing our work, as well as for your encouraging and positive feedback. Below, we address all your questions and concerns. If any issues remain, please let us know, and we’ll be happy to provide further clarification.
>
> **Novelty of the algorithm**
>
> We agree that the alternating minimization methods and its various formulations, such as EM-algorithm, have a long and rich history, which we also mention in our paper. At the same time, we believe that our optimization formulation for a particular MoE problem along with a particular open-sourced implementation (Algorithm 1 on page 11) is substantially novel. We provide it with extensive experiments, showing effective balance between general and personalized knowledge. Thank you for providing an additional reference, which we are happy to add to our work.
>
> **Violation of Assumption 1**
>
> We would like to note that our theoretical analysis demonstrates that the method is expected to converge quickly, at least under favorable conditions, a property that we believe is naturally expected from a well-designed method. However, we do not claim that this theory accounts for all possible practical scenarios. The primary validation of our algorithm for language modelling is provided through extensive experiments in Section 4.
>
> For over-parameterized models (which may include LLMs), it is possible that different $f_{valid}$ and $f_{train}$ share the same optima, which we also note in our paper (Lines 209-212, right). At the same time, as you also highlighted in your review, we see experimentally that the method performs favorably even for different distributions from the training datasets. We leave it as an interesting open question for further research: what are the general and joint conditions on $f_{valid}$ and $f_{train}$ that imply Assumption 1. Thanks.
>
>
> **Proof of strong convexity**
>
> Thank you for pointing this out. In the final version of our paper, we include the formal proof of strong convexity for our decoupled objective from Section F.3. Below, we provide a brief overview of our reasoning.
>
> Our goal is to show that the following function (see eq. (26) on page 27),
> $F(\Theta, \Phi, \Lambda) = l( \langle \lambda, \Theta^{\top} x \rangle) - \mu \langle \lambda, \Theta^{\top}x \rangle + \frac{\alpha}{2}( \| \Theta \|_F^2 + \| \Phi \|_F^2  ) + \mu d(\lambda) + \mu s(\Phi^{\top} x )$
> is strongly convex, for sufficiently large $\alpha > 0$ and $\mu > 0$.
>
> We notice that we can separate our objective in $\Phi$ and $\Theta$, thus constructing $g_1(\Theta, \Lambda) = l(\langle \Lambda, \Theta^\top x\rangle) + \frac{\alpha}{4}\|\Theta\|_F^2 +\frac{\mu}{2}d(\Lambda)$ and $g_2(\Phi, \Lambda) = -\mu \langle \Lambda, \Phi^\top x\rangle + \frac{\alpha}{4}\|\Phi\|_F^2 +\frac{\mu}{2}d(\Lambda)$, so that $F(\Theta, \Phi, \Lambda) = g_1(\Theta, \Lambda) + g_2(\Phi, \Lambda) + [\frac{\alpha}{4} (\|\Phi\|_F^2+\|\Theta\|_F^2) + \frac{\mu}{2}  d(\Lambda) + \mu s(\Phi^\top x)]$.
>
> Since the log-sum-exp function $s(\cdot)$ is strictly convex, while the negative entropy $d(\cdot)$ is strongly convex, as well as the Frobenius norm, we have the term in square brackets being strongly convex. Therefore, it suffices to prove the convexity for $g_1$ and $g_2$. For that, we compute the Hessian directly and show that it is positive semidefinite, for sufficiently large regularization parameters.
>
> For the formal proof, please also see Proposition 1 in the following link, which we will include in our updated appendix: https://anonymous.4open.science/r/CoMiGS/Strongly_Convex_Proof.pdf
>
>
> **Minor typos**
>
> Thanks for pointing out the typos and dimensions of the simplex, we will make sure these are fixed.

---

### Official Review · Reviewer_iYSM · 2025-03-08

**Overall Recommendation:** 3

**Summary:**

The authors focus on the problem of on-device collaborative fine-tuning of LLMs to address both computational resource heterogeneity and data heterogeneity among users. The authors try to develop a framework that can balance general and personalized knowledge for each token generation while being robust against overfitting. The experimental results show the proposed method improves the inference performance and computational efficiency.

**Claims And Evidence:**

Yes. The claims are easy to follow and the evidence is clearly supported.

**Essential References Not Discussed:**

No. I think the references are adequately covered.

**Ethical Review Concerns:**

N/A.

**Experimental Designs Or Analyses:**

Yes. The experiments are correctly configured and the insights obtained from the experiments are clearly explained.

**Methods And Evaluation Criteria:**

Yes. The authors use typical LLM models and tasks.

**Other Comments Or Suggestions:**

Overall, this paper is interesting and the technical depth is fine in most aspects.

**Other Strengths And Weaknesses:**

The authors propose separating the experts into generalists and specialists, where generalists are shared across users to provide general knowledge and specialists are localized to provide personalized knowledge. This is a practical idea to guarantee the model robustness.

**Questions For Authors:**

The authors propose to use a learnable router to determine the aggregation weights of the experts based on the input tokens. Could you please give more details on how to train this router, especially when the data is unbalanced among the users?

**Relation To Broader Scientific Literature:**

This paper is strongly related to the on-device LLM model design and deployment.

**Theoretical Claims:**

Yes. The formulation of distributed model training is clear.

---

> ### Author Rebuttal · Authors · 2025-04-01
>
> Thank you very much for your time and expertise in reviewing our work, as well as for your encouraging and positive feedback. Below, we address all your questions and concerns. If any issues remain, please let us know, and we’ll be happy to provide further clarification.
>
> Regarding your question about **router training**, we make the following clarifications:
> Within each user, the router training is the same as in standard MoE training, i.e. updated using gradient methods, apart from the following changes: 1) instead of updating router and expert parameters at the same time, we update the two sets of parameters in an alternating fashion. 2) router parameters are updated less frequently than expert parameters.
>
> Our method seems to handle **unbalanced data** very well. For example in Section 4.4, to simulate high and low local data quantities, we assigned 10x tokens to data to French and Italian users as to German and Dutch users (Line 387-389). Our results show that our method is robust to the local data imbalance, no matter whether local resource abundance is positively (Figure 5) or negatively (Figure 6) correlated with local data quantities.

---

### Official Review · Reviewer_RnkV · 2025-03-16

**Overall Recommendation:** 3

**Summary:**

This paper introduces CoMiGS, a modular federated learning framework for adapting LLMs using a mixture of generalist and specialist LoRA experts. CoMiGS employs a bi-level optimization strategy, alternating between routing and expert parameter updates. Experimental results on GPT-125M and Llama-3.2-1B demonstrate its superior performance over both local and federated baselines, while also offering theoretical and empirical insights into the framework’s behavior.

**Claims And Evidence:**

The paper makes a compelling case for the separation of global and user-specific parameters in a modular architecture that is federatedly trained. This enables specializing experts towards specific sub-distributions and efficient learning under non-IID data and heterogeneous target devices.

However, while the authors claim enabling on-device training, which is a typical scenario of cross-device federated settings, they do not evaluate in either case. The evaluation measures cost in a device-independent way, while the number of participating clients and participation scheme does not resemble cross-device settings.

**Essential References Not Discussed:**

-

**Experimental Designs Or Analyses:**

For the experimental evaluation, I have the following comments:

* It is unclear from the paper how the authors have federated the datasets and whether the distribution is non-IID amongst clients.
* Are the expert scores in Figure 4 the results of Eq.1?
* In §4.3, where the authors evaluate the adaptation to system heterogeneity in clients, an alternative could be not to have experts (or the same number of experts) across layers, especially given the dynamics of expert selection over the depth of the network.

**Methods And Evaluation Criteria:**

The proposed methods seem well motivated and the evaluation is reasonable. If anything, I would propose some additional areas of exploration to showcase the generality, scalability and applicability of the method. Concretely:

* Although perplexity gives a feel on the quality of the output, the number is not definitive on the downstream quality of the model. Towards this setting, it might be beneficial to also have LLM-as-judge reports on the models, or incorporate tasks like QA to understand the Question-Answering capabilities of the models.
* How would CoMiGS work with other PEFT methods or adapters (e.g. DORA, VERA)?
* How would CoMiGS work under different aggregation methods?
* Another interesting avenue for exploration, especially for on-device deployment, would be the interplay of the technique with quantization methods (or other compression schemes), where the router and adapters may operate on a lossy pretrained model.
* Since the paper inherits a federated setup, an interesting question arise wrt the tradeoff of utility and privacy when training the generalists under DP.

**Other Comments Or Suggestions:**

* The figures in the evaluation are barely legible. The authors should probably use a larger font size to increase legibility.
* In Table 2, the numbers in the parentheses should be explained in the caption.

**Other Strengths And Weaknesses:**

### Strengths

* The approach of having two sets of experts that are federated trained to specialize and route between global vs. local objectives is well motivated and work competitively to the baselines.
* The mixture of generalists and specialists allows the model to simultaneously capture individual user preferences and linguistic styles while still allowing exchange of globally shared knowledge. I like the interplay between generalists and specialists, especially wrt regularization.
* The evaluation explores various dimensions of the learning dynamics and modular behavior during inference, which provides interpretability in the operating dynamics of the model. I particularly like the fact that the authors have evaluated both in and out of distribution datasets.

### Weaknesses

* The federated paradigm put forward does not seem to be focusing on cross-device setup, but rather to assume small client sizes and full participation. Furthermore, the models have not been deployed "on-device".
* The dependence on validation sets may cause lack of robustness in the technique under distribution shifts.
* There seems to be much sensitivity of the results to the number of experts and data size. Although evaluated, there seems to be no proposed way to pick them, give a specific learning scenario.

**Questions For Authors:**

* I am not sure I have fully understood the reasoning behind the size of the trainable parameters (i.e. in the router) and the ability to update less frequently. Also, is the only way to achieve this by skipping updates in the outer loop, or can be effectively achieved by adjusting the learning rate of the outer optimization?
* Have the authors tested against also specializing the router function per user instead of federating it?
* Does Figure 3 suggest that we might not need the same number of specialists across layers? If so, is there the potential for further optimization?
* Following up from the results of Figure 5, what is a good way for practitioners to select how many specialists to dedicate to their learning model?

**Relation To Broader Scientific Literature:**

CoMiGS builds on top of the fields of federated learning, LLM adaptation and modular networks, by providing a framework that enables the personalization of on-device experts (specialists), while sharing knowledge across users via federated experts (generalists). It does so by taking into consideration both data and system heterogeneity, typical features of federated setups, thus adapting to non-IID multi-device federated environments.

**Theoretical Claims:**

I went over the theoretical alternative minimization convergence proofs in the appendix at a high-level. Looks reasonable.

---

> ### Author Rebuttal · Authors · 2025-04-01
>
> Thank you very much for your time and expertise in reviewing our work, as well as for your encouraging and positive feedback. Below, we address all your questions and concerns. If any issues remain, please let us know, and we’ll be happy to provide further clarification.
>
> **Methods And Evaluation Criteria**
>
> _Choice of perplexity as the metric_: First, we focus on next-word prediction for end users, and perplexity is a well-established metric for evaluating the quality of such predictions. Second, we believe that LLM-as-judge evaluations and existing QA tasks may not be the most appropriate metrics for assessing personalization, as the benchmarks we reviewed are primarily designed to test knowledge understanding. To properly evaluate personalization, we would need to curate custom personalized QA questions. However, since we are not experts in benchmark creation, we have chosen to rely on more standard evaluation metrics.
>
> **Experimental Designs Or Analyses**
>
> _Dataset distribution across users_: We briefly mentioned our user-specific dataset creation in Lines 259-260 -  A distinct category is assigned to a user, as it simulates the most challenging scenario for collaboration. For example, with Multi-lingual Wikipedia dataset, we assign each category (in this case, a language) to each of the four users.  The tokens per user are specified in Table 4 in the appendix. The data distribution is thus Non-IID, in the most extreme case.
>
> _Expert scores_: Yes you are right. Expert scores in Figure 4 are the results of Eq.1, we will make it clear in the updated manuscript. Eq.1 gives expert scores for each single token, and what we report in Figure 4 has been averaged across all tokens and multiple batches.
>
> _Number of specialists per layer_: This is a great question. Indeed we investigated the scenario where some users may not have experts, i.e. they are only equipped with generalists. Please check Figure 7 for this setting. Our experimental result shows that those low-resource users can still benefit from collaborating with high-resource users.
>
> **Weakness**
>
> _Cross-device setting?_ Due to academic budget constraints, we were unable to conduct cross-device experiments. Our current setup involves a single user equipped with a model containing hundreds of millions to billions of parameters, and we were unable to scale testing to multiple users. Nevertheless, we believe our framework remains applicable in cross-device scenarios.
> One can envision a hierarchical structure in which neighboring devices form local clusters that apply our method independently. The same approach can then be extended across clusters to enable broader coordination. Importantly, our method does not rely on full user participation—while full participation can accelerate the learning of a generalist model, it is not a requirement for the method to function effectively.
>
>
> _Dependence on valid set._ “The dependence on validation sets may cause lack of robustness in the technique under distribution shifts.” – Quite the opposite, for any OOD target distribution, as long as there is a small valid set, our router can learn a set of task-specific weights to combine generalists and specialists. Thus, CoMiGS gives great flexibility to tackle OOD target scenarios.
>
> _Sensitivity of CoMiGS_. Such sensitivity is expected—it would be surprising if a more capable device performed identically to a less capable one. We would like to emphasize that our method effectively disentangles resource availability from data quantity. As demonstrated in Figures 5 and 6, we evaluate two contrasting scenarios: one where low-data-quality users are assigned more experts and high-data-quality users fewer, and another with the reverse setup. In both cases, our CoMiGS method exhibits strong robustness.
>
>
> **Questions**
>
> _Router update and personalization:_ We followed standard MoE architecture, so the router is simply a one-layer MLP, that gives weights for each expert. You are right that avoiding router overfitting can as well be achieved with a smaller learning rate used for router update. In our approach, we went for less frequent updates, as it is more resource-efficient.
> Sorry for not making it clear, but our routers are localized per user, instead of being federated, as illustrated in Figure 2.
>
> _Futher optimization regarding #specialists per layer_: Indeed, Figure 3 suggests that different layers may not require the same number of experts. However, since each user exhibits a unique pattern of expert utilization across layers, it becomes challenging to perform further optimization at the level of individual users.
>
> _Suggestions to practitioners._ For practitioners, when there is no prior information on the local task complexity, we would suggest they select as many specialists as their devices allow. Our framework can effectively mitigate overfitting. In case the local task is complex, having more specialists can help with better performances, as illustrated in Figure 5.

---

### Decision · Program_Chairs · 2025-05-01

**Decision:**

Accept (poster)

**Comment:**

This paper studies a new Federated Learning setup to address both computational resource heterogeneity and data heterogeneity among users. The proposed approach uses a mixture of generalist and specialist LoRA experts. The method is justified both theoretically and empirically. All the concerns from the reviewers were addressed in the rebuttal.